# Hyperactivity of mTORC1- and mTORC2-dependent signaling mediates epilepsy downstream of somatic PTEN loss

Erin R Cullen[1†], Mona Safari[2,3], Isabelle Mittelstadt[1], Matthew C Weston[1,2,4*]

[1]Department of Neurological Sciences, Larner College of Medicine, University of Vermont, Burlington, United States; [2]Fralin Biomedical Research Institute at VTC, Center for Neurobiology Research, Roanoke, United States; [3]Translational Biology, Medicine, and Health Graduate Program, Roanoke, United States; [4]School of Neuroscience, Virginia Polytechnic and State University, Blacksburg, United States

*For correspondence:
mcweston7c@vt.edu

Present address: [†]Columbia University Irving Medical Center, Hammer Health Sciences Building, New York, United States

Competing interest: The authors declare that no competing interests exist.

**Abstract** Gene variants that hyperactivate PI3K-mTOR signaling in the brain lead to epilepsy and cortical malformations in humans. Some gene variants associated with these pathologies only hyperactivate mTORC1, but others, such as *PTEN*, *PIK3CA*, and *AKT*, hyperactivate both mTORC1- and mTORC2-dependent signaling. Previous work established a key role for mTORC1 hyperactivity in mTORopathies, however, whether mTORC2 hyperactivity contributes is not clear. To test this, we inactivated mTORC1 and/or mTORC2 downstream of early *Pten* deletion in a new mouse model of somatic *Pten* loss-of-function (LOF) in the cortex and hippocampus. Spontaneous seizures and epileptiform activity persisted despite mTORC1 or mTORC2 inactivation alone, but inactivating both mTORC1 and mTORC2 simultaneously normalized brain activity. These results suggest that hyperactivity of both mTORC1 and mTORC2 can cause epilepsy, and that targeted therapies should aim to reduce activity of both complexes.

## eLife assessment

This study investigated the role of specific proteins in a mouse model of developmental epilepsy. The significance of the work is **important** because a new mouse model was used to simulate a type of developmental epilepsy. The work is also significant because the deletion of two proteins together, but not separately, improved the symptoms, and data were **convincing**.

## Introduction

Genetic disruptions that hyperactivate mTOR activity, termed 'mTORopathies', cause macrocephaly, intractable childhood epilepsies, and behavioral presentations including autism spectrum disorders (*Crino, 2011*; *Lipton and Sahin, 2014*; *Costa-Mattioli and Monteggia, 2013*; *Wong and Crino, 2012*). The mTOR kinase can function in two complexes, mTORC1 and mTORC2, each with unique downstream targets and upstream activators (*Laplante and Sabatini, 2012*). A unifying feature of mTORopathies is hyperactivation of mTORC1, which has led to the hypothesis that this complex and its effectors mediate disease phenotypes (*Crino, 2020*). A subset of mTORopathies however (e.g. those caused by variants in *PTEN*, *PIK3CA*, and *AKT*) hyperactivate both mTORC1 and mTORC2 (*Dentel et al., 2019*; *Jansen et al., 2015*), and mTORC2 hyperactivity has also been reported in human TLE (*Talos et al., 2018*). It is unclear whether mTORC2 hyperactivity contributes to, or can cause, disease phenotypes such as epilepsy.

*Pten* loss-of-function (LOF) in mouse neurons induces soma hypertrophy, increased dendritic branching, and synaptic hyperconnectivity (*Weston et al., 2012*; *Williams et al., 2015*). These phenotypes resemble those observed in models of mTORC1-specific hyperactivation (*Nguyen and Bordey, 2021*), and can be prevented with either the mTORC1 inhibitor rapamycin or loss of the mTORC1-specific protein RAPTOR (*Tariq et al., 2022*). Rapamycin also rescues epilepsy and mortality associated with *Pten* loss, even after epilepsy has been established, suggesting that mTORC1 hyperactivity underlies these phenotypes (*Ljungberg et al., 2009*; *Nguyen et al., 2015*; *Sunnen et al., 2011*). Thus, there is strong evidence that hyperactivation of mTORC1 downstream of PTEN disruption causes the macrocephaly, epilepsy, early mortality, and synaptic dysregulation observed in humans and model organisms (*Zhou et al., 2009*).

There is also evidence against the mTORC1-centric hypothesis. Rapamycin may affect mTORC2 activity at high doses or over long periods of time (*Sarbassov et al., 2006*; *Zhou et al., 2009*), and mTORC2 regulates synapse function, neuronal size, and cytoskeletal organization independent of mTORC1 (*Angliker and Rüegg, 2013*; *McCabe et al., 2020*). In a mouse model in which *Pten* was inactivated in forebrain neurons in early adulthood, mTORC2 inhibition, but not mTORC1 inhibition, rescued seizures and behavioral abnormalities. In this model, epilepsy was dissociated from macrocephaly, which was rescued by mTORC1 inhibition but not mTORC2 inhibition (*Chen et al., 2019*). However, mTORC2 inactivation did not normalize spontaneous seizures or seizure susceptibility in a model of *Pten* loss in dentate granule neurons (*Cullen et al., 2023*). Thus, there is conflicting evidence about whether mTORC1 or mTORC2 hyperactivity causes epilepsy downstream of *Pten* loss.

To address this, we suppressed mTORC1 and mTORC2 activity alongside *Pten* loss by inducing simultaneous deletion of *Rptor* or *Rictor*, whose protein products are unique and essential components of mTORC1 and mTORC2, respectively, in a model of somatic *Pten* LOF. In this model, epilepsy could be rescued by concurrent mTORC1 and mTORC2 inactivation, but persisted when either gene remained intact, suggesting that hyperactivity of either complex can lead to neuronal hyperexcitability and epilepsy.

## Results

### Generation of a developmental brain somatic mosaic model of Pten LOF

To model the epileptogenic somatic mutations often observed in mTORopathies (*Kim and Lee, 2019*; *Koboldt et al., 2021*), we injected an AAV9 virus expressing green fluorescent protein (GFP) and Cre under control of the *SYN1* promoter into one hemisphere of the cortex of $Pten^{fl/fl}$, $Pten^{fl/fl}$-$Rptor^{fl/fl}$, $Pten^{fl/fl}$-$Rictor^{fl/fl}$, and $Pten^{fl/fl}$-$Rptor^{fl/fl}$-$Rictor^{fl/fl}$ mice at P0 (*Figure 1A*). We hereafter refer to these mice as Pten LOF, Pten-Rap LOF, Pten-Ric LOF, and PtRapRic LOF. GFP was expressed in neurons throughout the cortical layers in all experimental animals, but largely confined to one cortical hemisphere and the underlying hippocampus (*Figure 1B* and *Figure 1—figure supplement 1*). The percentage of GFP+ cells was similar across groups, although there was a decrease in cell density in the Pten LOF and Pten-Ric LOF groups (*Chen et al., 2015*; *Figure 1C*).

We analyzed the impact of *Pten*, *Rptor*, and *Rictor* LOF on mTORC1 and mTORC2 activity in the affected hemisphere using quantitative immunohistochemistry. Phospho-S6 levels, which report mTORC1 activity through S6K1, were increased from Controls in Pten LOF and Pten-Ric LOF brains, but rescued in Pten-Rap LOF brains (*Figure 1D$_1$*). Phospho-Akt (S473) levels, which report mTORC2 activity, were increased in Pten LOF and Pten-Rap LOF but normalized in Pten-Ric LOF brains (*Figure 1D$_2$*). Levels of both pS6 and pAkt were not different from Controls in PtRapRic LOF brains. These results indicate that *Pten* LOF hyperactivates both mTORC1 and mTORC2 in this model, and that these increases in activity are independently rescued by mTORC1 and mTORC2 inactivation.

### mTORC1 and/or mTORC2 inactivation attenuates Pten LOF-induced macrocephaly, cellular overgrowth, and electrophysiological changes

Macrocephaly and neuronal hypertrophy are well-characterized sequelae of deficient PTEN signaling in patients and mouse models (*Lasarge and Danzer, 2014*; *Zahedi Abghari et al., 2019*). In mouse models of *Pten* LOF, these phenotypes can be rescued by rapamycin treatment or *Rptor* LOF (*Chen et al., 2019*; *Kwon et al., 2003*; *Zhou et al., 2009*; *Tariq et al., 2022*). Here, cortical thickness

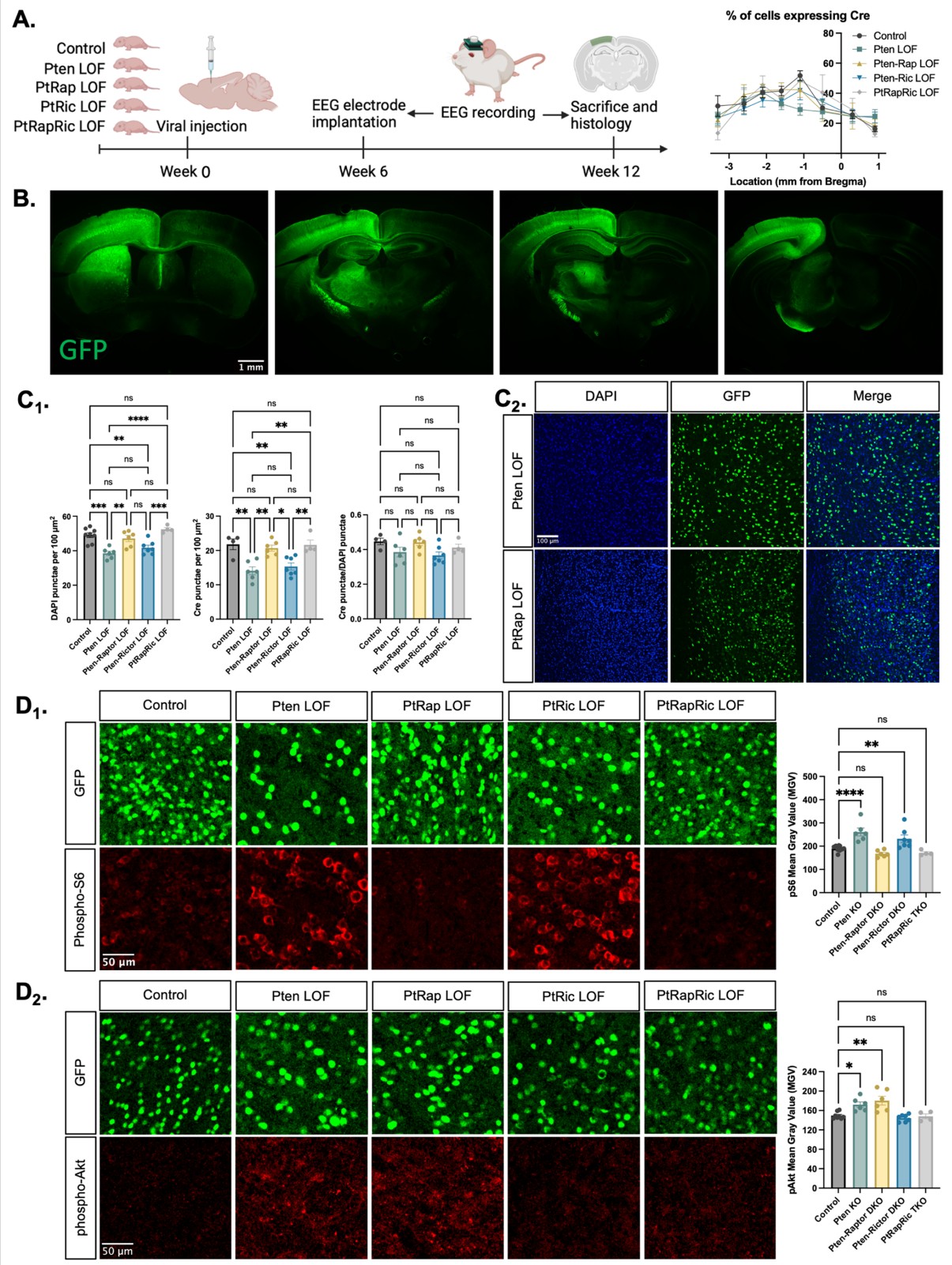

**Figure 1.** Histological characterization of the focal cortical Pten loss-of-function (LOF) model. (**A**) Experimental timeline showing induction of hSyn-Cre-GFP or Control hSyn-GFP AAV at P0 and EEG recording in adulthood. The location of cortical Cre expression relative to Bregma did not differ between groups. (**B**) Representative images of green fluorescent protein (GFP) expression in a Control mouse brain, demonstrating expression predominantly in one hemisphere of the cortex. Projections from affected neurons can be seen in white matter tracts including the fornix, internal capsule, corpus

*Figure 1 continued on next page*

*Figure 1 continued*

callosum, and cerebral peduncle. (**C1**) Quantification of lesion severity and neuron density in Cre-expressing animals. There were fewer Cre-expressing neurons per unit area in the Pten LOF and Pten-Ric LOF groups, which was at least partially attributable to a decrease in cell density in these groups. No significant differences in Cre expression remained when Cre expression was calculated based on cell density rather than area. (**C2**) Representative images of DAPI and Cre fluorescence in the cortex of Pten LOF and Pten-Rap LOF animals. Cell density and Cre density in Pten-Rap LOF animals were indistinguishable from Controls. (**D1**) Phospho-S6, a marker of mTORC1 activity, was increased by Pten LOF and reduced to control levels by concurrent *Rptor* loss. Phospho-S6 was also increased from Control levels in Pten-Ric LOF, indicating that mTORC1 hyperactivity was not normalized by *Rictor* loss. Combined *Rptor/Rictor* loss also normalized phospho-S6 expression. (**D2**) Phospho-Akt, a marker of mTORC2 activity, was increased in Pten LOF and normalized by *Rictor* loss, but not by *Rptor*. Combined *Rptor/Rictor* loss also normalized phospho-Akt expression. Error bars show mean ± SEM. ns indicates p>0.05, * indicates p<0.05, ** indicates p<0.01, *** indicates p<0.001, and **** indicates p<0.0001 as assessed by statistical tests indicated in *Table 3*. Panel A created with BioRender.com, and published using a CC BY-NC-ND license with permission.

The online version of this article includes the following source data and figure supplement(s) for figure 1:

**Source data 1.** File containing data used to generate graphs in *Figure 1*.

**Figure supplement 1.** Analysis of Cre expression in the hippocampus and its impact on outcome measures.

**Figure supplement 2.** Cre virus exposure does not significantly impact cortical morphology or baseline EEG.

was increased in Pten LOF mice but reduced to Control levels in Pten-Rap LOF, Pten-Ric LOF, and PtRapRic LOF mice (*Figure 2A–B*). In individual neurons, Pten LOF showed an ≈100% increase in mean soma size versus Control. Pten-Ric LOF significantly reduced soma size from Pten LOF levels, but still showed a 60% increase over Controls. Soma size in Pten-Rap LOF and PtRapRic LOF neurons did not significantly differ from Controls (*Figure 2C*). Whole-cell current-clamp analysis of neuronal membrane excitability of GFP+ neurons in acute brain slices supported these conclusions, as input resistance, capacitance, and rheobase were all significantly altered by Pten LOF, but not different from control levels in Pten-Rap LOF, Pten-Ric LOF, and PtRapRic LOF (*Figure 2D*). Spontaneous EPSCs (sEPSCs) were also recorded with whole-cell voltage clamp. Pten-Rap LOF and Pten-Ric LOF had elevated sEPSC frequency and amplitude, respectively, as previously reported (*Cullen et al., 2023*; *Chen et al., 2019*; *Tariq et al., 2022*). Neither of these parameters were different from Control in PtRapRic LOF neurons (*Figure 2D*), indicating that only concurrent mTORC1/mTORC2 inactivation can normalize sEPSC parameters.

## Concurrent mTORC1/2 inactivation, but neither alone, prevents epilepsy and interictal EEG abnormalities in focal Pten LOF

Next, we measured epileptic brain activity with video-EEG to determine the ability of mTORC1 or mTORC2 activity to rescue this key feature of mTORopathies. Generalized seizures (GS) were observed in 0/9 Control, 4/7 Pten LOF, 2/6 Pten-Rap LOF, 2/7 Pten-Ric LOF, and 0/6 PtRapRic LOF animals, suggesting that only simultaneous mTORC1 and mTORC2 inactivation potentially prevents GS. The frequency of GS events in Pten-Rap LOF and Pten-Ric LOF mice was not significantly lower than the GS frequency in Pten LOF mice, although the number of mice used was not powered to detect reductions in GS frequency. GS events were longer-lasting in Pten-Rap LOF animals than in Pten LOF or Pten-Ric LOF animals (*Figure 3A*), and did not appear to be correlated with mTOR pathway activity (*Figure 3—figure supplement 1*).

The most striking and consistent type of epileptic brain activity in the Pten LOF animals was frequent 5–7 Hz spike trains lasting 3 s or more, which fit previous characterizations of spike-and-wave discharges (SWDs) in rodents (*Cortez et al., 2004*). SWDs were observed in all Pten LOF, Pten-Rap LOF, and Pten-Ric LOF animals, and in 3/9 Control and 3/6 PtRapRic LOF animals. The frequency of SWDs in Pten LOF, Pten-Rap LOF, and Pten-Ric LOF animals was significantly higher than Controls, but PtRapRic LOF completely blocked this increase. Pten-Ric LOF animals had a lower frequency of SWD events than Pten LOF animals (*Figure 3B*). Taken together, these data indicate that inactivating mTORC1 provides no protection against *Pten* loss-induced epilepsy, and may even exacerbate it. mTORC2 inactivation provides some protection, but only concurrent mTORC1 and mTORC2 inactivation can prevent it.

In addition to epileptic activity, we also found that Pten LOF caused obvious alterations in features of the interictal EEG. To quantify these changes, we measured EEG coastline, absolute mean amplitude,

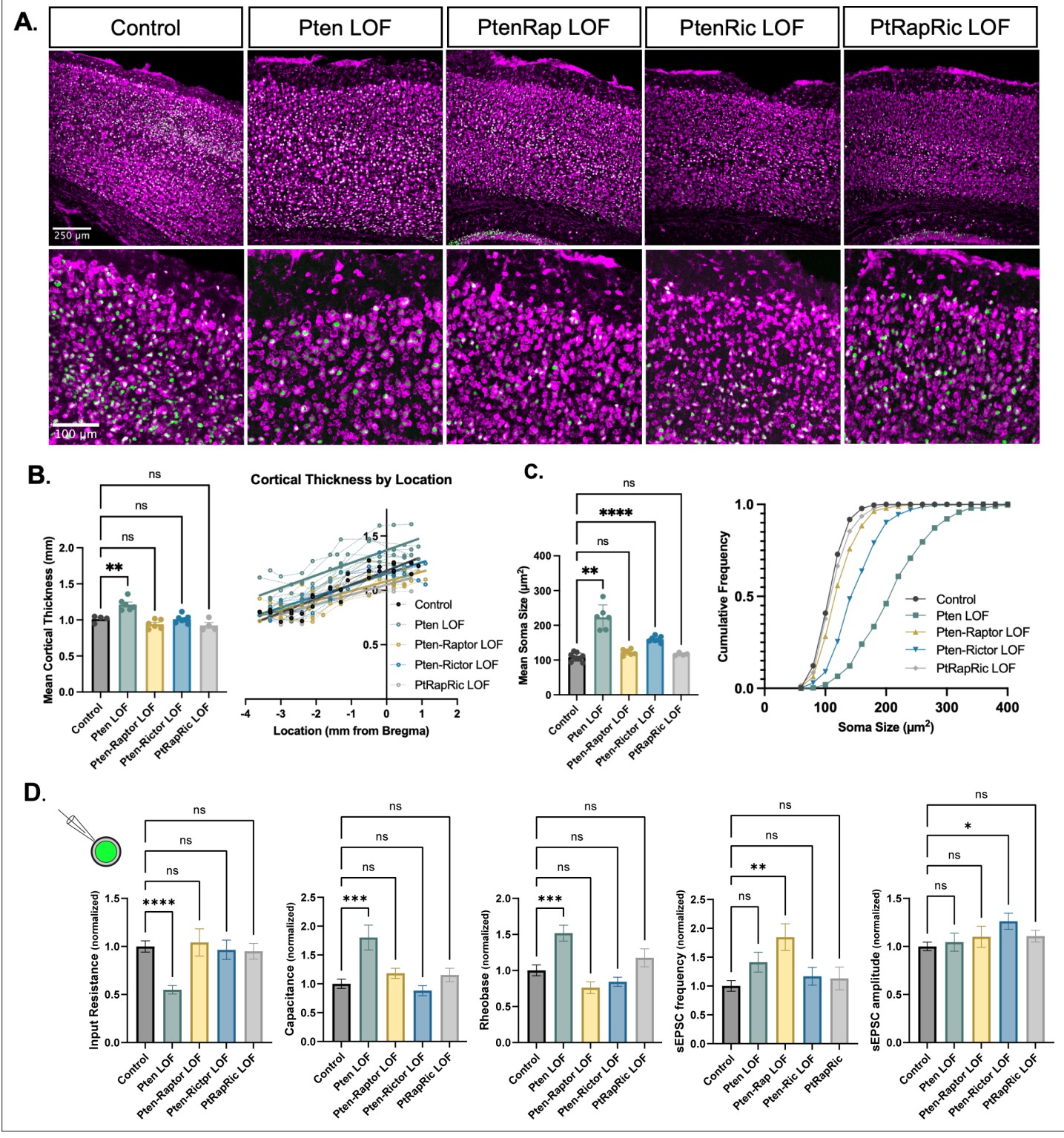

**Figure 2.** Independent mTORC1 or mTORC2 inactivation prevents most cellular effects of Pten loss-of-function (LOF), but only dual mTORC1/2 inactivation prevents all. (**A**) Example images showing fluorescent Nissl stain (magenta) and green fluorescent protein (GFP) expression (green) in cortical neurons. The top row shows the cortical thickness in all five groups and the bottom row shows zoomed-in images depicting the differences in soma size across groups. (**B**) The mean cortical thickness was increased in Pten LOF throughout the cortex. Pten-Rap, Pten-Ric, and PtRapRic LOF cortical thickness did not differ significantly from Controls. (**C**) The mean soma size was strongly increased in Pten LOF and to a smaller extent in Pten-Ric LOF. Pten-Rap LOF and PtRapRic LOF groups did not differ significantly from Controls. (**D**) Whole-cell patch clamp analysis of GFP+ neurons showed that

*Figure 2 continued on next page*

*Figure 2 continued*

capacitance and rheobase were increased in Pten LOF neurons, whereas input resistance was decreased. These changes were not found in in Pten-Rap LOF, Pten-Ric LOF, and PtRapRic LOF groups. The frequency of spontaneous EPSCs was elevated in Pten-Rap LOF neurons, and their amplitude was larger in Pten-Ric neurons. Patch data is normalized to values from littermate controls. Error bars show mean ± SEM. ns indicates p>0.05, * indicates p<0.05, ** indicates p<0.01, and **** indicates p<0.0001 as assessed by statistical tests indicated in *Table 3*.

The online version of this article includes the following source data for figure 2:

**Source data 1.** File containing raw data used to generate all graphs in *Figure 2*.

and power spectra of the EEG. Coastline, mean amplitude, and total power were all significantly increased by Pten LOF, and these changes were not significantly reduced by Pten-Rap or Pten-Ric LOF. PtRapRic LOF, however, reduced these increases to Control levels. Correspondingly, EEG power was increased in Pten LOF animals (*Figure 4A*; *Table 1A*). This increase was normalized by PtRapRic LOF, but not by Pten-Rap LOF or Pten-Ric LOF. PtRapRic LOF did not fully rescue a rightward shift in normalized EEG band power (*Figure 4B*; *Table 1B*).

## Discussion

Convergent discoveries have positioned mTORopathies as prime candidates for a precision medicine approach (*Moloney et al., 2021*; *Griffith and Wong, 2018*). There has been some clinical success with rapamycin analogues (*French et al., 2016*), but preclinical animal studies have suggested ways to further improve the selectivity and efficacy of the targets (*Nguyen and Bordey, 2021*). For the subset of mTORopathies in which mTORC1 but not mTORC2 signaling is hyperactive, selective inhibition of mTORC1 and its downstream effectors rescues many animal phenotypes, including seizures (*Karalis et al., 2022*; *Nguyen et al., 2022*). Studies using rapamycin as an mTORC1 inhibitor also came to this conclusion in *Pten* LOF models, but potential inhibition of mTORC2 and the finding that genetic inhibition of mTORC1 via *Rptor* deletion did not stop seizures, whereas inhibition of mTORC2 did (*Chen et al., 2019*) challenged this view. Here, we tested whether mTORC1 or mTORC2 inhibition alone were sufficient to block disease phenotypes in a model of somatic *Pten* LOF. Neither was, which agrees with previous findings that seizures persist after *Rptor* LOF and *Rictor* LOF in separate model systems (*Chen et al., 2019*; *Cullen et al., 2023*). This data could be interpreted as suggesting that epilepsy downstream of *Pten* LOF can proceed via mTORC-independent mechanisms, such as elevated β-catenin or its protein phosphatase activity (*Chen et al., 2015*; *Lyu et al., 2015*). However, we found that simultaneous mTORC1 and mTORC2 inhibition corrected all of the phenotypes of our model except for a rightward shift in relative EEG power (*Figure 4B*). This suggests that epilepsy can be caused by hyperactivity of either mTORC1- or mTORC2-dependent signaling downstream of *Pten* LOF, but not alternative pathways. Future studies should determine whether this is also true for other gene variants that hyperactivate both mTORC1 and mTORC2, as well in other models that have fully penetrant GS.

Epilepsy in Pten-Ric LOF mice likely emerges via mechanisms similar to those in mTOR pathway variants such as TSC that hyperactivate mTORC1 but not mTORC2, although additional effects of mTORC2 inactivation may contribute (*Cullen et al., 2023*). Epilepsy in Pten-Rap LOF occurs in the absence of macrocephaly, a hallmark of mTORC1 hyperactivity, possibly due to mTORC2 hyperactivity's effect on synaptic transmission (*McCabe et al., 2020*). Roles for both complexes in emergent neural network function are corroborated by findings that behavior can be altered by either mTORC1 or mTORC2 inactivation (*Angliker et al., 2015*). Rapamycin derivatives are somewhat selective for mTORC1 (*Choo and Blenis, 2009*; *Kim et al., 2002*) and are currently being tested in the clinic to treat mTORopathies and PTEN-related disorders (*Hardan et al., 2021*). Even more selective targeting of mTORC1 than can be achieved through rapamycin and its derivatives has been proposed as a way to effectively treat disease with fewer side effects (*Schreiber et al., 2019*). Our data suggest that this may only be true for mTORopathies that selectively hyperactive mTORC1. For others, including *PTEN*,

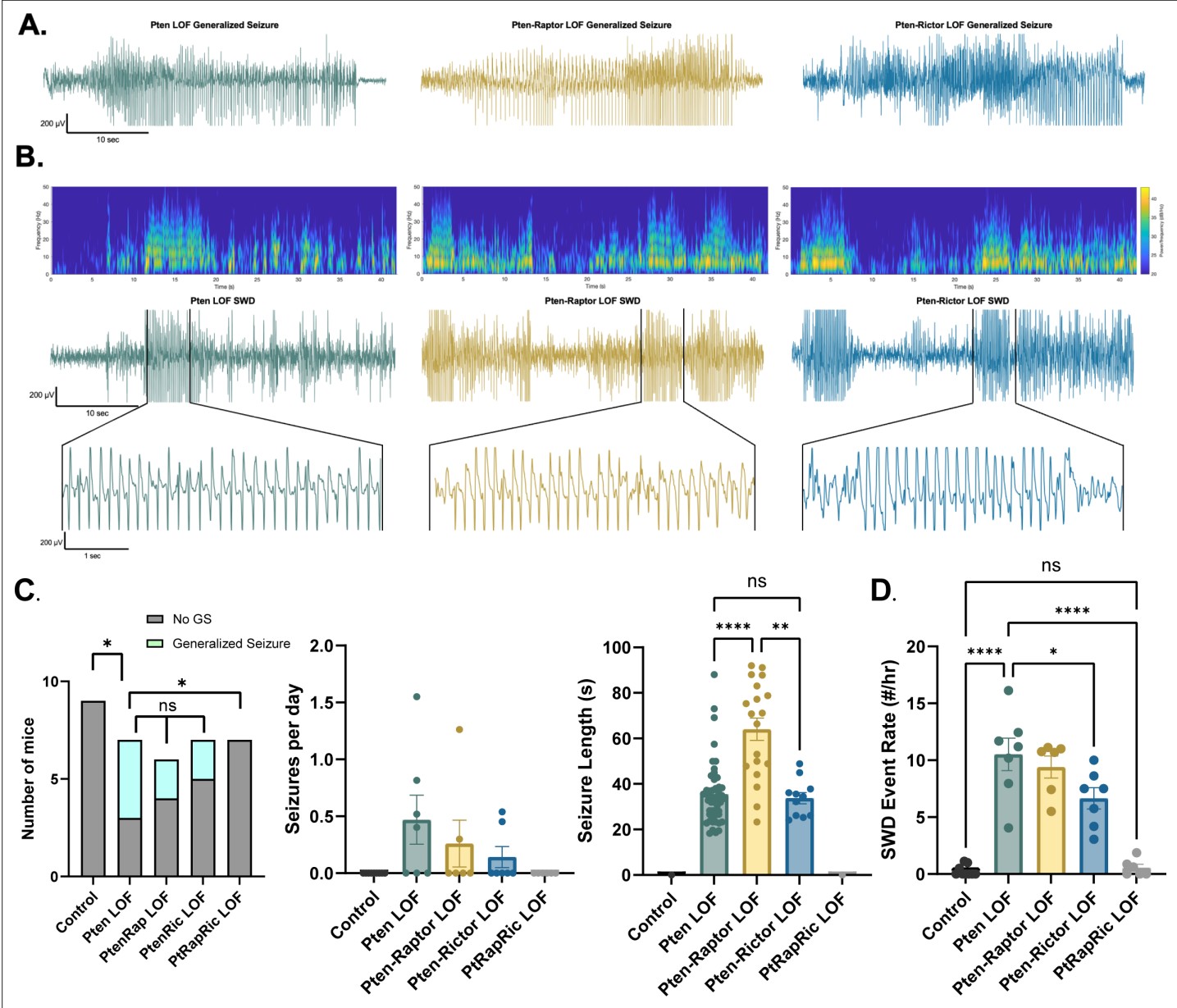

**Figure 3.** Combined mTORC1 and mTORC2 inactivation, but neither alone, rescues epilepsy in the focal Pten loss-of-function (LOF) model. Spontaneous seizures and interictal spike activity were assessed in Pten LOF, Pten-Rap LOF, Pten-Ric LOF, PtRapRic LOF, and Control mice. (**A**) Representative traces of generalized seizures (GS) in a subset of animals in Pten LOF, Pten-Rap LOF, and Pten-Ric LOF groups. (**B**) Spectrograms (top) and traces depicting example spike-and-wave discharge (SWD) event trains. (**C**) Summary data showing the occurrence of GS per animal, number of GS per day, and GS length. GS events were significantly longer in the Pten-Rap LOF group than the other groups. In the Pten LOF group, 19% of GS events (9/48) exceeded 45 s in length, and these events were observed in 2/4 Pten LOF GS+ animals. 79% of GS events (15/19) in the Pten-Rap LOF group exceeded 45 s, and these events were observed in 2/2 GS+ Pten-Rap LOF animals. 1/11 GS events in the Pten-Ric LOF group exceeded this threshold. (**D**) Summary data showing the SWD rate in all animals. Error bars show mean ± SEM. ns indicates p>0.05, * indicates p<0.05, ** indicates p<0.01, *** indicates p<0.001, and **** indicates p<0.0001 as assessed by tests indicated in *Table 3*.

The online version of this article includes the following source data and figure supplement(s) for figure 3:

**Source data 1.** File containing raw data used to generate graphs in *Figure 3*.

**Figure supplement 1.** Focal cortical Pten loss-of-function (LOF), Pten-Rap LOF, and Pten-Ric LOF cause a spectrum of outcomes.

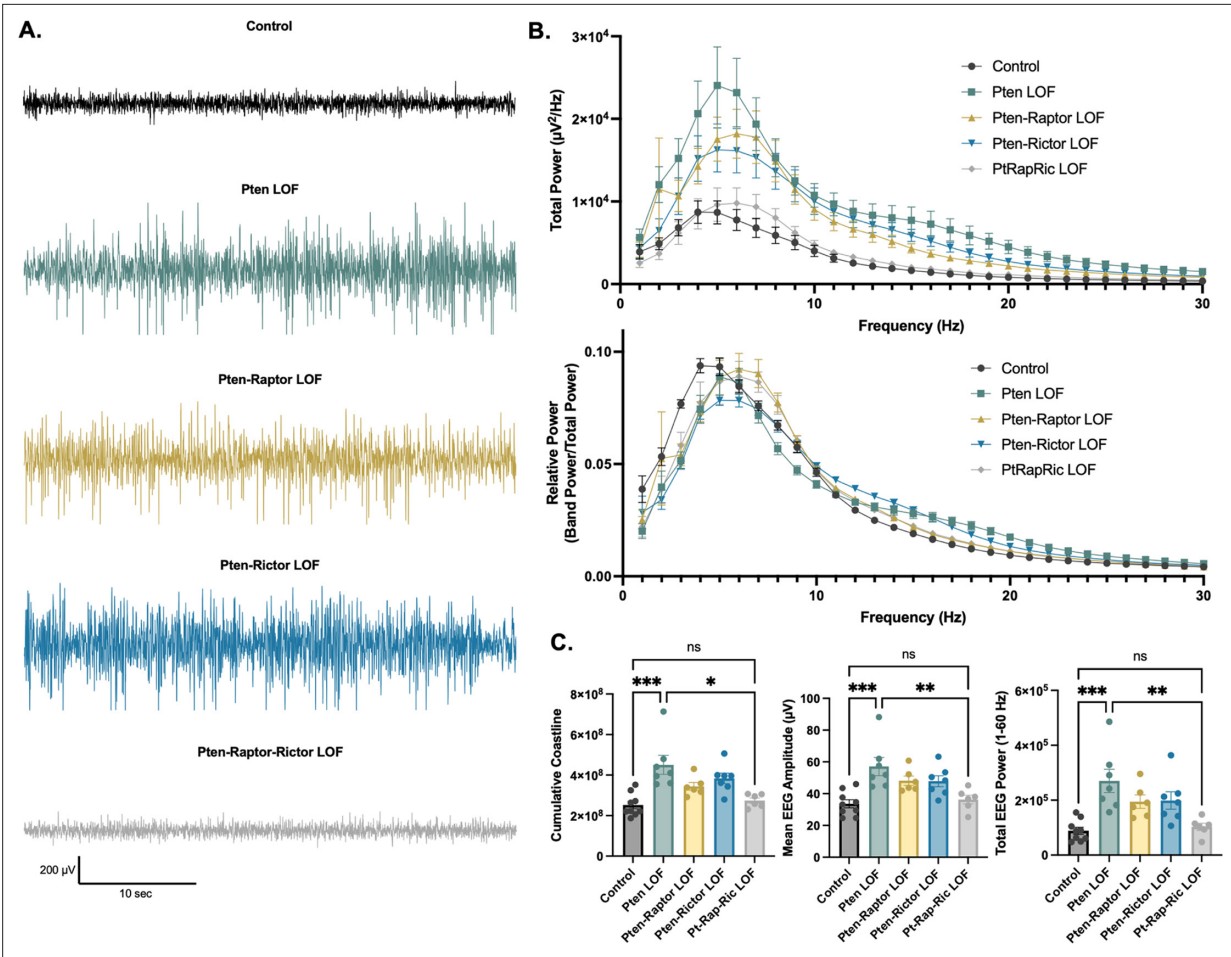

**Figure 4.** Combined mTORC1 and mTORC2 inactivation, but neither alone, rescues Pten loss-of-function (LOF)-induced abnormalities in the interictal EEG. (**A**) Examples of typical EEG traces for each genotype. In EEG epochs that were not characterized as GS or spike-and-wave discharge (SWD) events, Pten LOF animals had higher levels of EEG activity as quantified by EEG line length, absolute mean amplitude, and total power. These changes were not significantly decreased in Pten-Rap LOF or Pten-Ric LOF mice, but were normalized in PtRapRic LOF mice. (**B**) Total EEG power was increased by Pten LOF and attenuated, but not normalized, in either Pten-Rap or Pten-Ric LOF mice. Relative power was decreased in delta and increased in higher frequencies by Pten LOF. Pten-Rap LOF, Pten-Ric LOF, and PtRapRic LOF animals all showed a milder rightward shift of EEG power. (**C**) Line length, mean amplitude, and power are increased in Pten LOF and normalized by PtRapRic LOF. Error bars show mean ± SEM. ns indicates p>0.05, * indicates p<0.05, ** indicates p<0.01, *** indicates p<0.001 as assessed by statistical tests indicated in **Table 3**. Two-way ANOVA p-values for EEG power are reported in **Table 2**.

The online version of this article includes the following source data for figure 4:

**Source data 1.** File containing raw data used to generate graphs in **Figure 4**.

**Table 1.** EEG monitoring.

| Group | Mouse | Sex (M/F ratio) | Total hours | Mean rec. length | Mean rec. age (days) | GS rate | SWD rate |
|---|---|---|---|---|---|---|---|
| | 541 | F | 388.738 | 48.592 | 76.25 | 0 | 1.0 |
| | 833 | F | 237.772 | 47.554 | 62.2 | 0 | 0 |
| | 834 | F | 229.378 | 45.876 | 66.8 | 0 | 0 |
| | 841 | F | 312.545 | 62.509 | 59.8 | 0 | 0.5 |
| | 844 | M | 218.905 | 43.781 | 61.2 | 0 | 0 |
| | 157 | F | 166.736 | 27.789 | 73.2 | 0 | 0 |
| | 162 | M | 147.056 | 36.764 | 72.25 | 0 | 0 |
| Littermate Control (Pten$^{fl/fl}$; Control AAV) | Mean | 0.29 | 243.019 | 44.695 | 67.714 | 0 | 0.214 |
| | 570 | F | 160.217 | 16.022 | 57 | 0 | 1.125 |
| | 571 | M | 207.417 | 29.631 | 58.429 | 0 | 0 |
| C57/B6 Control (Cre AAV) | Mean | 0.5 | 183.817 | 22.826 | 57.714 | 0 | 0.5625 |
| | 576 | M | 265.171 | 44.195 | 88.333 | 0.402 | 11.688 |
| | 662 | F | 412.196 | 41.220 | 64.2 | 0.815 | 16.125 |
| | 677 | F | 173.978 | 28.996 | 60.667 | 0 | 10.188 |
| | 681 | M | 324.945 | 64.989 | 71.6 | 1.551 | 4.0625 |
| | 832 | M | 416.008 | 69.335 | 70 | 0.519 | 11.188 |
| | 842 | F | 260.195 | 28.9106 | 59.333 | 0 | 12.438 |
| | 867 | M | 427.933 | 42.793 | 59 | 0 | 7.964 |
| Pten$^{fl/fl}$;Raptor$^{+/+}$; Rictor$^{+/+}$ (Pten LOF) | Mean | 0.57 | 325.775 | 45.777 | 67.590 | 0.470 | 10.522 |
| | 538 | M | 438.481 | 48.720 | 75.667 | 0 | 11.125 |
| | 664 | M | 266.227 | 33.278 | 73.25 | 1.262 | 10.688 |
| | 665 | M | 400.567 | 57.224 | 72.429 | 0.299 | 11 |
| | 666 | M | 389.0367 | 64.839 | 67.167 | 0 | 10.75 |
| | 943 | M | 312.383 | 34.709 | 52.667 | 0 | 7.417 |
| | 999 | M | 272.4 | 30.267 | 60.556 | 0 | 5.5 |
| Pten$^{fl/fl}$;Raptor$^{fl/fl}$; Rictor$^{+/+}$ (Pten-Raptor LOF) | Mean | 0 | 346.516 | 44.840 | 66.956 | 0.260 | 9.413 |
| | 574 | F | 319.483 | 63.897 | 61.8 | 0 | 10 |
| | 676 | F | 284.317 | 40.617 | 66 | 0 | 7.5 |
| | 678 | F | 238.584 | 23.858 | 85.6 | 0.453 | 4.1875 |
| | 811 | M | 307.438 | 61.488 | 64 | 0 | 3.0625 |
| | 941 | M | 295.691 | 59.138 | 65.6 | 0 | 5.75 |
| | 41 | F | 397.683 | 36.153 | 64.364 | 0 | 8.625 |
| | 42 | M | 421.85 | 42.185 | 61.1 | 0.554 | 7.5 |
| Pten$^{fl/fl}$;Raptor$^{+/+}$; Rictor$^{fl/fl}$ (Pten-Rictor LOF) | Mean | 0.43 | 323.578 | 46.762 | 66.923 | 0.144 | 6.661 |

*Table 1 continued on next page*

*Table 1 continued*

| Group | Mouse | Sex (M/F ratio) | Total hours | Mean rec. length | Mean rec. age (days) | GS rate | SWD rate |
|---|---|---|---|---|---|---|---|
| | 933 | M | 286.467 | 31.830 | 58.778 | 0 | 1.875 |
| | 940 | M | 234.533 | 78.178 | 52.667 | 0 | 0.625 |
| | 997 | M | 209.497 | 19.045 | 54.909 | 0 | 0 |
| | 40 | F | 220.717 | 18.393 | 59.5 | 0 | 0.875 |
| | 158 | F | 154.228 | 20.705 | 73.2 | 0 | 0 |
| Pten^fl/fl;Raptor^fl/fl; Rictor^fl/fl (Pten-Raptor-Rictor LOF) | 161 | M | 168.006 | 56.002 | 72 | 0 | 0 |
| | Mean | 0.66 | 203.750 | 34.152 | 61.842 | 0 | 0.5625 |

*PIK3CA*, and *AKT*, dual inhibitors or inhibitors of PI3K, which are being tested preclinically (**Theilmann et al., 2020**; **Roy et al., 2015**; **White et al., 2020**), are likely to be required.

# Materials and methods

## Key resources table

| Reagent type (species) or resource | Designation | Source or reference | Identifiers | Additional information |
|---|---|---|---|---|
| Genetic reagent (*Mus musculus*) | *Rptor*-cKO mice | The Jackson Laboratories | Jackson Labs stock: 013188 | |
| Genetic reagent (*Mus musculus*) | *Rictor*-cKO mice | The Jackson Laboratories | Jackson Labs stock: 020649 | |
| Genetic reagent (*Mus musculus*) | *Pten*-cKO mice | The Jackson Laboratories | Jackson Labs stock: 006440 | |
| Strain, strain background (*Mus musculus*) | C57BL/6J | The Jackson Laboratories | Jackson Labs stock: 006440 | |
| Recombinant DNA reagent | AAV9-*hSYN*-GFP-Cre | Addgene | Addgene_105540 | |
| Recombinant DNA reagent | AAV9-*hSYN*-GFP | Addgene | Addgene_105539 | |
| Antibody | Guinea pig polyclonal anti-NeuN | Synaptic Systems | Cat #266 004 | 1:1000 |
| Antibody | Rabbit monoclonal anti-pAkt (S473) | Cell Signaling | Cat #4060 RRID:AB_2315049 | 1:1000 |
| Antibody | Rabbit monoclonal anti-pS6 (S240/244) | Cell Signaling | Cat #5364 | 1:1000 |
| Other | Nissl stain | Invitrogen | Invitrogen N21482 | 1:50 |
| Software, algorithm | pClamp | Molecular Devices | RRID:SCR_011323 | |
| Software, algorithm | Sirenia | Pinnacle | RRID:SCR_016183 | |
| Software, algorithm | AxoGraph X | AxoGraph | RRID:SCR_014284 | |
| Software, algorithm | SPSS | SPSS | RRID:SCR_002865 | |
| Software, algorithm | Prism | GraphPad | RRID: SCR_002798 | |
| Software, algorithm | Fiji | NIH | RRID:SCR_002285 | |
| Software, algorithm | MATLAB | Mathworks | RRID:SCR_001622 | |

## Animals

This study was performed in strict accordance with the recommendations in the Guide for the Care and Use of Laboratory Animals of the National Institutes of Health. All surgery was performed under isofluorane anesthesia, and every effort was made to minimize suffering. All mice were bred, and procedures were conducted at the Fralin Biomedical Research Institute at VTC, or at the University of Vermont. Each institution is fully accredited by the Association for Assessment and Accreditation

**Table 2.** EEG power.

**Total power**

| Band | Two-way ANOVA p-value | | | | | | |
|---|---|---|---|---|---|---|---|
| | Control vs. Pten LOF | Control vs. Pten-Raptor LOF | Control vs. Pten-Rictor LOF | Control vs. Pten-Raptor-Rictor LOF | Pten LOF vs. Pten-Raptor LOF | Pten LOF vs. Pten-Rictor LOF | Pten LOF vs. Pten-Raptor-Rictor LOF |
| 1 | 0.7215 | 0.9996 | 0.9976 | 0.7021 | 0.8249 | 0.9347 | 0.1485 |
| 2 | 0.0833 | 0.8162 | 0.8509 | 0.7749 | >0.9999 | 0.276 | 0.043 |
| 3 | 0.0657 | 0.4328 | 0.5564 | 0.9949 | 0.5893 | 0.6387 | 0.0543 |
| 4 | 0.1205 | 0.2658 | 0.3091 | >0.9999 | 0.6358 | 0.79 | 0.1219 |
| 5 | 0.0834 | 0.1019 | 0.1838 | 0.9946 | 0.7452 | 0.6116 | 0.1144 |
| 6 | 0.0508 | 0.0779 | 0.127 | 0.8868 | 0.8606 | 0.6302 | 0.0988 |
| 7 | 0.0395 | 0.0829 | 0.0875 | 0.6727 | 0.9963 | 0.8508 | 0.1137 |
| 8 | 0.0321 | 0.0749 | 0.0746 | 0.6353 | >0.9999 | 0.9834 | 0.1093 |
| 9 | 0.0236 | 0.0727 | 0.0766 | 0.8311 | 0.9923 | 0.9993 | 0.0576 |
| 10 | 0.0156 | 0.0605 | 0.0609 | 0.9112 | 0.9096 | 0.9974 | 0.0297 |
| 11 | 0.0148 | 0.0469 | 0.0336 | 0.8046 | 0.7577 | 0.9889 | 0.0275 |
| 12 | 0.017 | 0.0376 | 0.0186 | 0.5816 | 0.7035 | 0.9837 | 0.0321 |
| 13 | 0.0219 | 0.0431 | 0.0138 | 0.5078 | 0.6552 | 0.9572 | 0.0379 |
| 14 | 0.0322 | 0.0298 | 0.0109 | 0.5927 | 0.4738 | 0.9128 | 0.0486 |
| 15 | 0.047 | 0.0172 | 0.0087 | 0.7629 | 0.35 | 0.8344 | 0.0616 |
| 16 | 0.0497 | 0.0133 | 0.0081 | 0.5221 | 0.2767 | 0.7401 | 0.0669 |
| 17 | 0.0386 | 0.0092 | 0.0104 | 0.4807 | 0.2108 | 0.6195 | 0.0524 |
| 18 | 0.0284 | 0.0104 | 0.0129 | 0.4624 | 0.1732 | 0.4864 | 0.0386 |
| 19 | 0.0241 | 0.0166 | 0.0139 | 0.607 | 0.1619 | 0.3975 | 0.0315 |
| 20 | 0.0221 | 0.0132 | 0.0179 | 0.3517 | 0.1556 | 0.368 | 0.0342 |
| 21 | 0.021 | 0.008 | 0.0176 | 0.3236 | 0.1487 | 0.3703 | 0.0334 |
| 22 | 0.0238 | 0.0078 | 0.0188 | 0.3161 | 0.1752 | 0.4039 | 0.0382 |
| 23 | 0.0279 | 0.0093 | 0.021 | 0.3728 | 0.2164 | 0.4387 | 0.0434 |
| 24 | 0.0303 | 0.0131 | 0.0247 | 0.4069 | 0.2433 | 0.4609 | 0.0463 |
| 25 | 0.0314 | 0.0191 | 0.0253 | 0.3983 | 0.2521 | 0.4648 | 0.0479 |
| 26 | 0.0366 | 0.0144 | 0.026 | 0.4339 | 0.2631 | 0.4714 | 0.0552 |
| 27 | 0.0399 | 0.0079 | 0.0264 | 0.4669 | 0.2576 | 0.4914 | 0.0599 |
| 28 | 0.0463 | 0.0064 | 0.0273 | 0.4853 | 0.2697 | 0.5059 | 0.0687 |
| 29 | 0.0559 | 0.0122 | 0.0282 | 0.4671 | 0.3075 | 0.5461 | 0.0831 |
| 30 | 0.0598 | 0.0141 | 0.0287 | 0.4514 | 0.3209 | 0.5616 | 0.0894 |

**Relative power**

| Band | Two-way ANOVA p-value | | | | | | |
|---|---|---|---|---|---|---|---|
| | Control vs. Pten LOF | Control vs. Pten-Raptor LOF | Control vs. Pten-Rictor LOF | Control vs. Pten-Raptor-Rictor LOF | Pten LOF vs. Pten-Raptor LOF | Pten LOF vs. Pten-Rictor LOF | Pten LOF vs. Pten-Raptor-Rictor LOF |
| 1 | <0.0001 | <0.0001 | 0.0033 | <0.0001 | 0.4967 | 0.0572 | 0.9743 |
| 2 | <0.0001 | 0.9986 | <0.0001 | <0.0001 | 0.0009 | 0.3614 | 0.9977 |
| 3 | <0.0001 | <0.0001 | <0.0001 | <0.0001 | 0.9337 | 0.9777 | 0.2279 |

*Table 2 continued on next page*

*Table 2 continued*

**Total power**

| | | | | | | | |
|---|---|---|---|---|---|---|---|
| 4 | <0.0001 | <0.0001 | <0.0001 | <0.0001 | 0.997 | 0.8997 | 0.9044 |
| 5 | 0.5938 | 0.4988 | <0.0001 | 0.1852 | >0.9999 | 0.0063 | 0.9593 |
| 6 | 0.9826 | 0.0834 | 0.189 | 0.5724 | 0.3384 | 0.0796 | 0.9046 |
| 7 | 0.6153 | <0.0001 | 0.9878 | 0.0043 | <0.0001 | 0.8905 | <0.0001 |
| 8 | 0.0053 | 0.0074 | >0.9999 | 0.0208 | <0.0001 | 0.0114 | <0.0001 |
| 9 | 0.0065 | 0.9007 | 0.9998 | 0.832 | 0.0006 | 0.0053 | 0.0003 |
| 10 | 0.3806 | 0.9964 | 0.8873 | 0.9994 | 0.2612 | 0.0714 | 0.3248 |
| 11 | >0.9999 | 0.8412 | 0.1377 | 0.9282 | 0.9154 | 0.2466 | 0.9685 |
| 12 | 0.7573 | 0.4594 | 0.0076 | 0.6587 | 0.9923 | 0.2905 | 0.9999 |
| 13 | 0.2732 | 0.3278 | 0.002 | 0.5321 | >0.9999 | 0.5486 | 0.994 |
| 14 | 0.086 | 0.6136 | 0.0015 | 0.6451 | 0.8414 | 0.8216 | 0.819 |
| 15 | 0.0278 | 0.8819 | 0.0033 | 0.7984 | 0.3346 | 0.9894 | 0.4322 |
| 16 | 0.0089 | 0.9533 | 0.0105 | 0.9064 | 0.1119 | 0.9998 | 0.1569 |
| 17 | 0.0059 | 0.9659 | 0.0539 | 0.9184 | 0.0744 | 0.9312 | 0.1137 |
| 18 | 0.0062 | 0.9594 | 0.1831 | 0.9362 | 0.0828 | 0.7147 | 0.1027 |
| 19 | 0.0162 | 0.9653 | 0.4248 | 0.9619 | 0.1467 | 0.6116 | 0.152 |
| 20 | 0.0594 | 0.9772 | 0.6562 | 0.9777 | 0.2931 | 0.6739 | 0.2915 |
| 21 | 0.1876 | 0.9896 | 0.8068 | 0.9836 | 0.5003 | 0.8099 | 0.5378 |
| 22 | 0.397 | 0.9948 | 0.9024 | 0.9905 | 0.706 | 0.9022 | 0.7437 |
| 23 | 0.5863 | 0.996 | 0.9476 | 0.9958 | 0.8445 | 0.9485 | 0.8465 |
| 24 | 0.7443 | 0.998 | 0.972 | 0.998 | 0.9163 | 0.975 | 0.9174 |
| 25 | 0.8341 | 0.9992 | 0.9844 | 0.9988 | 0.9447 | 0.9851 | 0.9517 |
| 26 | 0.8939 | 0.9997 | 0.9931 | 0.9992 | 0.9617 | 0.9887 | 0.972 |
| 27 | 0.9375 | >0.9999 | 0.9965 | 0.9995 | 0.9721 | 0.9934 | 0.9851 |
| 28 | 0.9652 | >0.9999 | 0.9981 | 0.9996 | 0.9815 | 0.9967 | 0.993 |
| 29 | 0.9801 | >0.9999 | 0.9988 | 0.9996 | 0.9879 | 0.9983 | 0.9972 |
| 30 | 0.9886 | >0.9999 | 0.9993 | 0.9995 | 0.9922 | 0.9991 | 0.9991 |

of Laboratory Animal Care, and all protocols were approved by their respective Institutional Animal Care and Use Committees. All experiments were performed in accordance with respective state and federal Animal Welfare Acts and the policies of the Public Health Service. The animal protocol numbers for Virginia Tech were: 22-198 and 22-130, and the University of Vermont were: 16-001 and 19-034. *Rptor* (Jackson Labs #013188) and *Rictor* (Jackson Labs #020649) homozygous floxed mice were crossed to a *Pten* homozygous floxed line (***Groszer et al., 2001***, Jackson Labs #006440). Experimental animals were the offspring of two animals homozygous floxed for *Pten* and heterozygous floxed for *Raptor* and/or *Rictor*. Intracranial viral injections were done at P0 or P1. Animals that were homozygous for the floxed *Pten* allele and either homozygous or wild-type at the *Raptor* and/or *Rictor* allele were injected with an AAV9 viral vector expressing an eGFP-Cre fusion driven by the *hSyn* promoter (Addgene #105540). Control animals were of the same genotype but were injected with an AAV9 virus expressing eGFP under control of the *hSyn* promoter, but not Cre (Addgene #105539). Pup injections were conducted with hypothermic anesthesia. A Nanoject (Drummond) was used to deliver 100 nl of Cre or control virus to three sites spanning the left hemisphere of the cortex. Pups were quickly rewarmed and returned to their home cage. Additional control animals with no

floxed genes on a C57B6/J background (Jackson Labs #000644) were injected with the eGFP-Cre expressing AAV9 to ensure that Cre expression did not have effects independent of the *Pten^{fl/fl}* genotype (*Figure 1—figure supplement 2*).

## Surgeries and EEG recordings

Animals were aged to 6 weeks and then implanted with a wireless EEG monitoring headcap. Animals were anesthetized with 4% isofluorane and arranged on a stereotaxic surgical apparatus with ear bars. Anesthesia was maintained with 1.5–2.5% isofluorane. The skull surface was exposed and holes were made with a 23 g needle at six locations on the skull surface. 3/32" screws (Antrin Miniature Specialties) were inserted into the skull to record epidural EEG. Screws were secured with VetBond then covered with surgical cement. Recording leads were placed bilaterally at approximately Bregma AP +1.0 ML 1.0, AP −1.0 ML 1.5, and AP −2.5 ML 1.5. The screws were attached to a six-pin Millimax strip which was secured to the skull with additional cement. Animals were administered 5 mg/kg ketoprofen post surgery and allowed to recover for at least 5 days before EEG recording. Surgical protocol was based on guidance provided by Pinnacle Technologies (https://www.pinnaclet.com).

After recovery from surgery, animals were fitted with wireless three-channel EEG preamplifiers (Pinnacle Technologies) and EEG signal was recorded at 256 Hz with simultaneous video recording. EEG recording session length was dependent on battery life. A total of at least 150 hr of EEG data was collected for each experimental animal (*Table 1*).

## Histology and imaging

EEG-implanted animals were euthanized by isofluorane overdose followed by transcardiac perfusion with 4% paraformaldehyde. Brains were postfixed in 4% paraformaldehyde for 24 hr and then transferred to 30% sucrose for at least 48 hr. 40 µm coronal slices were cut and preserved in a cryoprotectant solution (50% PBS, 30% ethylene glycol, 20% glycerol). Prior to staining, slices were washed three times with PBS. Eight unstained slices spanning the affected area (Bregma −3.6 through Bregma +1.1) were mounted with DAPI Fluoromount-G (SouthernBiotech) and used to assess cortical thickness, GFP expression, and cell density. GFP expression, cell density, and soma size were measured in the upper layers of the cortex to reduce within-subject variability. GFP expression was also measured in the dentate gyrus, CA1, and CA3 regions of the dorsal hippocampus. For other analyses, three sections per animal at approximately Bregma −1.6, −2.1, and −2.6 were used. A fluorescent Nissl stain (Invitrogen N21482, 1:50) was used to assess soma size.

After washing, slices were placed in blocking solution (10% normal goat serum, 0.1% Triton X-100, and PBS) for 1 hr. Slices were incubated in the following primary antibodies for 3 hr at room temperature: phospho-S6 Ribosomal Protein Ser240/244 (rabbit monoclonal, 1:1000 dilution, Cell Signaling Technology, catalog #5364, RRID: AB_10694233), phospho-AKT Ser473 (rabbit monoclonal, 1:1000 dilution, Cell Signaling Technology, catalog #4060), and NeuN (guinea pig polyclonal, 1:1000 dilution, Synaptic Systems, catalog #266004). Following primary antibody application, slices were washed three times in PBS and incubated in Alexa Fluor secondary antibodies goat anti-guinea pig 647 and goat anti-rabbit 594 (Invitrogen) for 1 hr at room temperature.

Slides stained for Nissl substance, NeuN/pS6, and NeuN/pAkt were imaged at 10× (2048×2048) with 5 µm z-step on a Nikon C2 confocal microscope (UVM Microscopy Imaging Core). One image of the left (GFP-expressing) cortical hemisphere in each of three region-matched slices per animal was collected for analysis. Widefield images (2560×2160) of unstained DAPI-mounted cross-sectional slices were taken with a Zyla sCMOS camera (Andor) mounted on an upright microscope (IX73; Olympus) at 5× and 20× resolution. Pixel width was manually calculated using an 0.1 mm hemocytometer.

## Slice electrophysiology

Slice electrophysiology was conducted at P14–30. Animals were deeply anesthetized and decapitated, and the brain was quickly dissected into ice-cold cutting solution (126 mM NaCl, 25 mM NaHCO$_3$, 10 mM d-glucose, 3.5 mM KCl, 1.5 mM NaH$_2$PO$_4$, 0.5 mM CaCl$_2$, 10.0 mM MgCl$_2$, pH 7.3–7.4). 350 µm slices were cut using a Leica 1000S Vibratome. Slices were transferred to 37°C aCSF (126 mM NaCl, 3.5 mM KCl, 1.0 mM MgCl$_2$, 2.0 mM CaCl$_2$, 1.5 mM NaH$_2$PO$_4$, 25 mM NaHCO$_3$, and 10 mM d-glucose, pH 7.3–7.4) and incubated for 30 min. The slices were then incubated at room temperature for at least another 30 min prior to recording. All solutions were continuously bubbled with 95% O$_2$ and 5% CO$_2$.

**Table 3.** Results of statistical tests.

**Cell density (DAPI) | Classic ANOVA | F=12.44, p<0.0001**

| Group | Control | | *Pten* LOF | | *PtenRap* LOF | | *PtenRic* LOF | | *PtRapRic* LOF | |
|---|---|---|---|---|---|---|---|---|---|---|
| Mean ± SEM | 49.09±1.45 | | 38.52±1.28 | | 47.1±1.83 | | 41.78±1.44 | | 52.49±0.984 | |

| Comparison | Con-Pten | Con-PRa | Con-PRi | Con-PRaRi | Pten-PRa | Pten-PRi | Pten-PRaRi | Pra-PRi | Pra-PRaRi | Pri-PRaRi |
|---|---|---|---|---|---|---|---|---|---|---|
| p-Value | 0.0002 | 0.8627 | 0.0072 | 0.5859 | 0.0044 | 0.5357 | <0.0001 | 0.111 | 0.2043 | 0.001 |

**Cre density | Classic ANOVA | F=10.15, p<0.0001**

| Group | Control | | *Pten* LOF | | *PtenRap* LOF | | *PtenRic* LOF | | *PtRapRic* LOF | |
|---|---|---|---|---|---|---|---|---|---|---|
| Mean ± SEM | 21.79±1.56 | | 14.08±1.15 | | 20.69±0.876 | | 15.33±1.03 | | 21.62±1.41 | |

| Comparison | Con-Pten | Con-PRa | Con-PRi | Con-PRaRi | Pten-PRa | Pten-PRi | Pten-PRaRi | Pra-PRi | Pra-PRaRi | Pri-PRaRi |
|---|---|---|---|---|---|---|---|---|---|---|
| p-Value | 0.0018 | 0.9684 | 0.0074 | >0.9999 | 0.0027 | 0.9181 | 0.0022 | 0.0131 | 0.9827 | 0.0093 |

**Cre/DAPI ratio | Classic ANOVA | F=2.616, p=0.0629**

| Group | Control | | *Pten* LOF | | *PtenRap* LOF | | *PtenRic* LOF | | *PtRapRic* LOF | |
|---|---|---|---|---|---|---|---|---|---|---|
| Mean ± SEM | 0.447±0.0190 | | 0.385±0.0280 | | 0.441±0.0184 | | 0.366±0.0194 | | 0.411±0.0196 | |

| Comparison | Con-Pten | Con-PRa | Con-PRi | Con-PRaRi | Pten-PRa | Pten-PRi | Pten-PRaRi | Pra-PRi | Pra-PRaRi | Pri-PRaRi |
|---|---|---|---|---|---|---|---|---|---|---|
| p-Value | 0.3742 | 0.9998 | 0.1294 | 0.8646 | 0.3568 | 0.9622 | 0.9336 | 0.1022 | 0.8935 | 0.6387 |

**Phospho-S6 MGV | Classic ANOVA | F=11.69, p<0.0001**

| Group | Control | | *Pten* LOF | | *PtenRap* LOF | | *PtenRic* LOF | | *PtRapRic* LOF | |
|---|---|---|---|---|---|---|---|---|---|---|
| Mean ± SEM | 190±4.22 | | 261±17.0 | | 168±5.87 | | 232±16.2 | | 171±5.34 | |

| Comparison | Con-Pten | Con-PRa | Con-PRi | Con-PRaRi | Pten-PRa | Pten-PRi | Pten-PRaRi | Pra-PRi | Pra-PRaRi | Pri-PRaRi |
|---|---|---|---|---|---|---|---|---|---|---|
| p-Value | <0.0001 | 0.1569 | 0.0074 | 0.2862 | <0.0001 | 0.0809 | <0.0001 | 0.0004 | 0.8616 | 0.0023 |

**Phospho-Akt MGV | Kruskal-Wallis test | K=19.92, p=0.0006**

| Group | Control | | *Pten* LOF | | *PtenRap* LOF | | *PtenRic* LOF | | *PtRapRic* LOF | |
|---|---|---|---|---|---|---|---|---|---|---|
| Mean ± SEM | 149±2.46 | | 174±6.15 | | 177±8.10 | | 144±2.75 | | 148±4.86 | |

| Comparison | Con-Pten | Con-PRa | Con-PRi | Con-PRaRi | Pten-PRa | Pten-PRi | Pten-PRaRi | Pra-PRi | Pra-PRaRi | Pri-PRaRi |
|---|---|---|---|---|---|---|---|---|---|---|
| p-Value | 0.0126 | 0.0077 | 0.3828 | 0.7533 | 0.8739 | 0.0012 | 0.0171 | 0.0007 | 0.0115 | 0.6792 |

**Cortical thickness (μm) | Classic ANOVA | F=16.60, p<0.0001**

| Group | Control | | *Pten* LOF | | *PtenRap* LOF | | *PtenRic* LOF | | *PtRapRic* LOF | |
|---|---|---|---|---|---|---|---|---|---|---|
| Mean ± SEM | 1.01±0.0225 | | 1.21±0.0341 | | 0.940±0.0254 | | 1.01±0.0215 | | 0.924±0.0399 | |

| Comparison | Con-Pten | Con-PRa | Con-PRi | Con-PRaRi | Pten-PRa | Pten-PRi | Pten-PRaRi | Pra-PRi | Pra-PRaRi | Pri-PRaRi |
|---|---|---|---|---|---|---|---|---|---|---|
| p-Value | 0.0011 | 0.4682 | >0.9999 | 0.3622 | <0.0001 | 0.0002 | <0.0001 | 0.3519 | 0.9956 | 0.2723 |

**Soma size (μm²) | Welch's ANOVA | W=44.00, p<0.0001**

| Group | Control | | *Pten* LOF | | *PtenRap* LOF | | *PtenRic* LOF | | *PtRapRic* LOF | |
|---|---|---|---|---|---|---|---|---|---|---|
| Mean ± SEM | 109±3.88 | | 223±14.6 | | 123±3.48 | | 160±3.18 | | 117±1.90 | |

| Comparison | Con-Pten | Con-PRa | Con-PRi | Con-PRaRi | Pten-PRa | Pten-PRi | Pten-PRaRi | Pra-PRi | Pra-PRaRi | Pri-PRaRi |
|---|---|---|---|---|---|---|---|---|---|---|
| p-Value | 0.0021 | 0.1536 | <0.0001 | 0.5204 | 0.0041 | 0.0539 | 0.0055 | <0.0001 | 0.7598 | <0.0001 |

**GS occurrence (binary) | Generalized linear model | Wald = 15.13, p=0.004**

*Table 3 continued on next page*

*Table 3 continued*

**Cell density (DAPI) | Classic ANOVA | F=12.44, p<0.0001**

| Group | Control | | *Pten* LOF | | *PtenRap* LOF | *PtenRic* LOF | | *PtRapRic* LOF | |
|---|---|---|---|---|---|---|---|---|---|
| Mean ± SEM | 0/9 | | 4/7 | | 2/6 | 2/7 | | 0/6 | |
| Comparison | *Con-Pten* | *Con-PRa* | Con-PRi | *Con-PRaRi* | Pten-PRa | Pten-PRi | Pten-PRaRi | Pra-PRi | Pra-PRaRi | Pri-PRaRi |
| p-Value | 0.023 | 0.833 | .933 | 1 | 1 | 1 | 0.023 | 1 | 0.833 | 0.943 |

**GS frequency (#/day) | Kruskal-Wallis test | K=9.475, p=0.0503**

| Group | Control | | *Pten* LOF | | *PtenRap* LOF | *PtenRic* LOF | | *PtRapRic* LOF | |
|---|---|---|---|---|---|---|---|---|---|
| Mean ± SEM | 0.00±0.00 | | 0.470±0.216 | | 0.260±0.206 | 0.142±0.0920 | | 0.00±0.00 | |
| Comparison | *Con-Pten* | *Con-PRa* | Con-PRi | *Con-PRaRi* | Pten-PRa | Pten-PRi | Pten-PRaRi | Pra-PRi | Pra-PRaRi | Pri-PRaRi |
| p-Value | 0.0678 | >0.9999 | >0.9999 | >0.9999 | >0.9999 | >0.9999 | 0.1418 | >0.9999 | >0.9999 | >0.9999 |

**GS length (all events; s) | Kruskal-Wallis test | K=23.29, p<0.0001**

| Group | Control | | *Pten* LOF | | *PtenRap* LOF | *PtenRic* LOF | | *PtRapRic* LOF | |
|---|---|---|---|---|---|---|---|---|---|
| Mean ± SEM | n/a | | 35.5±2.02 | | 64.1±4.93 | 33.8±2.46 | | n/a | |
| Comparison | *Con-Pten* | *Con-PRa* | Con-PRi | *Con-PRaRi* | Pten-PRa | Pten-PRi | Pten-PRaRi | Pra-PRi | Pra-PRaRi | Pri-PRaRi |
| p-Value | n/a | n/a | n/a | n/a | <0.0001 | >0.9999 | n/a | 0.0021 | n/a | n/a |

**SWD frequency (#/hour) | Classic ANOVA | F=32.36, p<0.0001**

| Group | Control | | *Pten* LOF | | *PtenRap* LOF | *PtenRic* LOF | | *PtRapRic* LOF | |
|---|---|---|---|---|---|---|---|---|---|
| Mean ± SEM | 0.292±0.156 | | 10.5±1.42 | | 9.41±0.969 | 6.66±0.930 | | 0.563±0.304 | |
| Comparison | *Con-Pten* | *Con-PRa* | Con-PRi | *Con-PRaRi* | Pten-PRa | Pten-PRi | Pten-PRaRi | Pra-PRi | Pra-PRaRi | Pri-PRaRi |
| p-Value | <0.0001 | <0.0001 | <0.0001 | 0.9994 | 0.9025 | 0.0258 | <0.0001 | 0.214 | <0.0001 | 0.0003 |

**SWD length (animal mean; s) | Kruskal-Wallis test | K=2.420, p=0.7280**

| Group | Control | | *Pten* LOF | | *PtenRap* LOF | *PtenRic* LOF | | *PtRapRic* LOF | |
|---|---|---|---|---|---|---|---|---|---|
| Mean ± SEM | 5.33±1.59 | | 5.15±0.395 | | 5.36±0.245 | 5.11±0.283 | | 5.14±0.167 | |
| Comparison | *Con-Pten* | *Con-PRa* | Con-PRi | *Con-PRaRi* | Pten-PRa | Pten-PRi | Pten-PRaRi | Pra-PRi | Pra-PRaRi | Pri-PRaRi |
| p-Value | >0.9999 | >0.9999 | >0.9999 | >0.9999 | >0.9999 | >0.9999 | >0.9999 | >0.9999 | >0.9999 | >0.9999 |

**Cumulative coastline (line length*10[8]) | Kruskal-Wallis test | K=22.97, p=0.0001**

| Group | Control | | *Pten* LOF | | *PtenRap* LOF | *PtenRic* LOF | | *PtRapRic* LOF | |
|---|---|---|---|---|---|---|---|---|---|
| Mean ± SEM | 2.52±0.186 | | 4.51±0.469 | | 3.44±0.194 | 3.83±0.260 | | 2.74±0.133 | |
| Comparison | *Con-Pten* | *Con-PRa* | Con-PRi | *Con-PRaRi* | Pten-PRa | Pten-PRi | Pten-PRaRi | Pra-PRi | Pra-PRaRi | Pri-PRaRi |
| p-Value | 0.0005 | 0.2844 | 0.011 | >0.9999 | >0.9999 | >0.9999 | 0.0128 | >0.9999 | >0.9999 | 0.1179 |

**Mean EEG amplitude (µV) | Classic ANOVA | F=7.223, p=0.0003**

| Group | Control | | *Pten* LOF | | *PtenRap* LOF | *PtenRic* LOF | | *PtRapRic* LOF | |
|---|---|---|---|---|---|---|---|---|---|
| Mean ± SEM | 33.7±2.55 | | 57.1±5.71 | | 48.2±2.96 | 47.8±3.45 | | 36.3±2.78 | |
| Comparison | *Con-Pten* | *Con-PRa* | Con-PRi | *Con-PRaRi* | Pten-PRa | Pten-PRi | Pten-PRaRi | Pra-PRi | Pra-PRaRi | Pri-PRaRi |
| p-Value | 0.0004 | 0.0576 | 0.0505 | 0.9851 | 0.4792 | 0.4018 | 0.0049 | >0.9999 | 0.2388 | 0.2322 |

**Mean EEG power (µV$^2$*10$^5$) | Classic ANOVA | F=8.123, p=0.0001**

*Table 3 continued on next page*

*Table 3 continued*

**Cell density (DAPI) | Classic ANOVA | F=12.44, p<0.0001**

| Group | Control | *Pten* LOF | *PtenRap* LOF | *PtenRic* LOF | *PtRapRic* LOF | | | | |
|---|---|---|---|---|---|---|---|---|---|
| Mean ± SEM | 0.888±0.128 | 2.70±0.426 | 1.95±0.244 | 1.99±0.317 | 1.03±0.134 | | | | |

| Comparison | *Con-Pten* | *Con-PRa* | *Con-PRi* | *Con-PRaRi* | *Pten-PRa* | *Pten-PRi* | *Pten-PRaRi* | *Pra-PRi* | *Pra-PRaRi* | *Pri-PRaRi* |
|---|---|---|---|---|---|---|---|---|---|---|
| p-Value | 0.0002 | 0.0613 | 0.0349 | 0.9954 | 0.3373 | 0.3552 | 0.0019 | >0.9999 | 0.2003 | 0.14 |

Individual slices were transferred to a recording chamber located on an upright microscope (BX51; Olympus) and were perfused with heated (32–34°C), oxygenated aCSF (2 ml/min).

Whole-cell voltage-clamp and current-clamp recordings were obtained using Multiclamp 700B and Clampex 10.7 software (Molecular Devices). GFP+ cells in cortical layer 2/3 in the motor cortex were targeted for patching. Intracellular solution contained (in mM): 136 mM K-gluconate, 17.8 mM HEPES, 1 mM EGTA, 0.6 mM $MgCl_2$, 4 mM ATP, 0.3 mM GTP, 12 mM creatine phosphate, and 50 U/ml phosphocreatine kinase, pH 7.2. When patch electrodes were filled with intracellular solution, their resistance ranged from 4 to 6 MΩ. Access resistance was monitored continuously for each cell.

For current-clamp experiments, the intrinsic electrophysiological properties of neurons were tested by injecting 500 ms square current pulses incrementing in 20 pA steps, starting with –100 pA. The membrane time constant was calculated from an exponential fit of current stimulus offset. Input resistance was calculated from the steady state of the voltage responses to the hyperpolarizing current steps. Membrane capacitance was calculated by dividing the time constant by the input resistance. Action potentials (APs) were evoked with 0.5 s, 20 pA depolarizing current steps. Rheobase was defined as the minimum current required to evoke an AP during the 500 ms of sustained somatic current injection. For voltage-clamp experiments to measure sEPSC frequency and amplitude, neurons were held at –70 mV and recorded for 2 min. All electrophysiology data were analyzed offline with AxoGraph X software (AxoGraph Scientific).

## EEG analysis

EEG files were converted from Pinnacle's proprietary format to European Data Format (EDF) files and imported to MATLAB. A filtering program was used to flag traces in which seizures were likely to be occurring based on amplitude and line length between data points. For Control and PtRapRic LOF animals, all 10 s epochs in which signal amplitude exceeded 400 μV at two time points at least 1 s apart were manually reviewed by a rater blinded to genotype. This method was not suitable for the *Pten* LOF, *Pten-Raptor* LOF, and *Pten-Rictor* LOF animals because of the density of high-amplitude interictal activity they displayed. For these animals, at least 100 of the highest-amplitude traces were manually reviewed and then traces with persistent abnormally low amplitude, often indicating post-ictal suppression, were reviewed as well. Flagged traces were displayed for a rater to mark the beginning and end of each seizure. We also reviewed at least 48 hr of data from each animal manually. All seizures that were identified during manual review were also identified by the automated detection program. SWD events were manually marked in eight evenly spaced 1 hour epochs in each of two 24 hr recording sessions per animal and verified on video not to be caused by movement such as chewing or scratching.

Baseline EEG measures were taken from a representative sample of EEG files for each animal. 692 evenly spaced 5 second epochs were sampled over 24 hr, repeated for two recording sessions per animal. Line length was defined as the sum of the linear distances between adjacent data points during the 5 s analysis epoch. Mean amplitude was defined as the mean of the absolute value of data points in the 5 s analysis epoch. Power spectral density was calculated with the pwelch function in MATLAB. All three EEG channels were analyzed and the mean of all channels was used for statistical analysis.

## Image analysis

Image analysis was conducted using ImageJ/Fiji. Cell density and Cre expression in the cortex were automatically assessed using the Fiji Analyze Particles tool. Cre expression in the hippocampus was

manually assessed by counting Cre-expressing punctae within a 100 μm linear portion of the region of interest. Because neuronal somas within the hippocampal cell layers are so closely packed together, we were unable to resolve DAPI punctae to assess cell density in the hippocampus. Soma size was measured by dividing Nissl stain images into a 10 mm$^2$ grid. The somas of all GFP-expressing cells fully within three randomly selected grid squares in Layer II/III were manually traced. pS6 and pAkt expression were measured by drawing 200 μm wide columns spanning all cortical layers. Background was subtracted from images with a 30 μm rolling ball algorithm (Fiji/ImageJ). The mean pixel value of the column was recorded and values were averaged by animal.

## Statistical analysis

Prism 9 or 10 (GraphPad Prism) was used to conduct statistical analyses and create graphs. All data are presented as mean ± SEM. Power analyses on preliminary and published data were used to calculate the number of animals necessary for this study, with the exception of GS occurrence, which is discussed below. All data distributions were assessed for normality using the Shapiro-Wilk test. If data did not meet the criteria for normal distribution ($p<0.05$), Kruskal-Wallis tests were used to assess statistical relationships with Dunn's post hoc correction for multiple comparisons. Variance was assessed using Brown-Forsythe tests. If variance differed significantly ($p<0.05$), Welch's ANOVA test was used to assess statistical relationships. Group differences in normally distributed datasets with equal variances were assessed using one-way ANOVA with Tukey post hoc correction for multiple comparisons. Power spectral density was assessed using two-way ANOVA with Tukey post hoc analysis. Seizure occurrence was assessed with a generalized linear model with a binary distribution and logistic link function implemented in SPSS and Bonferroni corrected for multiple comparisons. See *Tables 2 and 3* for details regarding statistical tests. A power analysis showed that an excessive number of animals would be required to detect a decrease in GS incidence or frequency by *Rptor* or *Rictor* LOF in this model, thus we chose not to pursue the possibility that *Rptor* or *Rictor* loss decreases these outcome measures.

## Acknowledgements

Research reported in this publication was supported by NINDS grant R01NS110945 (MCW), as well as P20GM135007, Core C: Customized Physiology and Imaging Core. Images created with BioRender. We thank Caitlynn Barrows, Elise Prehoda, and Willie Tobin for early experiments helping create the mouse model.

## Additional information

### Funding

| Funder | Grant reference number | Author |
| --- | --- | --- |
| National Institute of Neurological Disorders and Stroke | NS110945 | Erin R Cullen |
| NINDS | R01NS110945 | Matthew C Weston |

The funders had no role in study design, data collection and interpretation, or the decision to submit the work for publication.

### Author contributions

Erin R Cullen, Formal analysis, Investigation, Writing – original draft, Project administration, Writing – review and editing; Mona Safari, Formal analysis, Investigation; Isabelle Mittelstadt, Investigation; Matthew C Weston, Conceptualization, Supervision, Funding acquisition, Writing – original draft, Project administration, Writing – review and editing

### Author ORCIDs

Erin R Cullen 🔗 http://orcid.org/0000-0002-0134-8717
Matthew C Weston 🔗 http://orcid.org/0000-0001-5558-7070

### Ethics

This study was performed in strict accordance with the recommendations in the Guide for the Care and Use of Laboratory Animals of the National Institutes of Health. All surgery was performed under isofluorane anesthesia, and every effort was made to minimize suffering. All mice were bred, and procedures were conducted at the Fralin Biomedical Research Institute at VTC, or at the University of Vermont. Each institution is fully accredited by the Association for Assessment and Accreditation of Laboratory Animal Care, and all protocols were approved by their respective Institutional Animal Care and Use Committees. All experiments were performed in accordance with respective state and federal Animal Welfare Acts and the policies of the Public Health Service. The animal protocol numbers for Virginia Tech were: 22-198 and 22-130, and the University of Vermont were: 16-001 and 19-034.

Reviewer #1 (Public Review): https://doi.org/10.7554/eLife.91323.3.sa1
Reviewer #2 (Public Review): https://doi.org/10.7554/eLife.91323.3.sa2
Reviewer #3 (Public Review): https://doi.org/10.7554/eLife.91323.3.sa3
Author Response https://doi.org/10.7554/eLife.91323.3.sa4

---

## Additional files

### Supplementary files

• MDAR checklist

### Data availability

All data generated or analyzed during this study are included in the manuscript and supporting source data files and tables.

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
