## [Editor Report · eLife assessment]

This study investigated the role of specific proteins in a mouse model of developmental epilepsy. The significance of the work is **important** because a new mouse model was used to simulate a type of developmental epilepsy. The work is also significant because the deletion of two proteins together, but not separately, improved the symptoms, and data were **convincing**.

---

## [Referee Report · Reviewer #1 (Public Review)]

Hyperactivation of mTOR signaling causes epilepsy. It has long been assumed that this occurs through overactivation of mTORC1, since treatment with the mTORC1 inhibitor rapamycin suppresses seizures in multiple animal models. However, the recent finding that genetic inhibition of mTORC1 via Raptor deletion did not stop seizures while inhibition of mTORC2 did, challenged this view (Chen et al, Nat Med, 2019). In the present study, the authors tested whether mTORC1 or mTORC2 inhibition alone was sufficient to block the disease phenotypes in a model of somatic Pten loss-of-function (a negative regulator of mTOR). They found that inactivation of either mTORC1 or mTORC2 alone normalized brain pathology but did not prevent seizures, whereas dual inactivation of mTORC1 and mTORC2 prevented seizures. As the functions of mTORC1 versus mTORC2 in epilepsy remain unclear, this study provides important insight into the roles of mTORC1 and mTORC2 in epilepsy caused by Pten loss and adds to the emerging body of evidence supporting a role for both complexes in the disease development.

Strengths:

The animal models and the experimental design employed in this study allow for a direct comparison between the effects of mTORC1, mTORC2, and mTORC1/mTORC2 inactivation (i.e., same animal background, same strategy and timing of gene inactivation, same brain region, etc.). Additionally, the conclusions on brain epileptic activity are supported by analysis of multiple EEG parameters, including seizure frequencies, sharp wave discharges, interictal spiking, and total power analyses.

Weaknesses:

The original concerns regarding the hippocampal contribution to the seizure phenotypes in this Pten loss-of-function model have been addressed with the inclusion of new data in the revised manuscript.

The issue of sample sizes being small and do not allow for the assessment of whether mTORC1 or mTORC2 inactivation reduces seizure frequency or incidence remains a limitation of the study. However, the study's main conclusion that spontaneous seizures and epileptiform activity persist following inactivation of mTORC1 or mTORC2 alone while it is rescued following inactivation of both mTORC1 and mTORC2 is supported by the provided data and remains valid.

---

## [Referee Report · Reviewer #2 (Public Review)]

Summary: The study by Cullen et al presents intriguing data regarding the contribution of mTOR complex 1 (mTORC1) versus mTORC2 or both in Pten-null induced macrocephaly and epileptiform activity. The role of mTORC2 in mTORopathies, and in particular Pten loss-off-function (LOF)-induced pathology and seizures, is understudied and controversial. In addition, recent data provided evidence against the role of mTORC1 in PtenLOF-induced seizures. To address these controversies and the contribution off these mTOR complexes in PtenLOF-induced pathology and seizures, the authors injected a AAV9-Cre into the cortex of conditional single, double and triple transgenic mice at postnatal day 0 to remove Pten, Pten+Raptor or Rictor, and Pten+raptor+rictor. Raptor and Rictor are essentially binding partners of mTORC1 and mTORC2, respectively. One major finding is that despite preventing the mild macrocephaly and increased cell size, Raptor knockout (KO, decrease mTORC1 activity) did not prevent the occurrence of seizures and the rate of SWD event, and aggravated seizure duration. Similarly, Rictor KO (decreased mTORC2 activity) partially prevented the mild macrocephaly and increased cell size but did not prevent the occurrence of seizures and did not affect seizure duration. However, Rictor KO reduced the rate of SWD events. Finally, the pathology and seizure/SWD activity were fully prevented in the double KO. These data suggest the contribution of both increased mTORC1 and mTORC2 in the pathology and epileptic activity of Pten LOF mice, emphasizing the importance of blocking both complexes for seizure treatment. Whether these data apply to other mTORopathies due to Tsc1, Tsc2, mTOR, AKT or other gene variants remain to be examined.

Strengths: The strengths are as follow: (1) they address an important and controversial question that has clinical application, (2) the study uses a reliable and relatively easy method to KO specific genes in cortical neurons, based on AAV9 injections in pups. (2) they perform careful video-EEG analyses correlated with some aspects of cellular pathology.

Weaknesses: the study has nevertheless a few weaknesses: (1) the conclusions are perhaps a bit overstated. The data do not show that increased mTORC1 or mTORC2 are sufficient to cause epilepsy. But the data clearly show that both increased mTORC1 and mTORC2 activity contribute to the pathology and seizure activity and as such are necessary for seizures to occur. (2) the data related to the EEG would benefit from having more mice. Adding more mice would have help determine whether there is a decrease in seizure activity with the Rictor or Raptor KO. (3) it would have been interesting to examine the impact of mTORC2 and mTORC1 overexpression related to point #1 above.

The authors properly addressed my comments. Number 3 above was only a suggestion that could be a follow-up in another study.

---

## [Referee Report · Reviewer #3 (Public Review)]

Summary: This study investigated the role of mTORC1 and 2 in a mouse model of developmental epilepsy which simulates the epilepsy in cortical malformations. Given activation of genes such as PTEN activate TORC1, and this is considered to be excessive in cortical malformations, the authors asked whether inactivating mTORC1 and 2 would ameliorate the seizures and malformation in the mouse model. The work is highly significant because a new mouse model is used where Raptor and Rictor, which regulate mTORC1 and 2 respectively, were inactivated in one hemisphere of the cortex. The work is also significant because the deletion of both Raptor and Rictor improved the epilepsy and malformation. In the mouse model, the seizures were generalized or there were spike wave discharges (SWD). They also examined the interictal EEG. The malformation was manifested by increased cortical thickness and soma size.

Strengths: The presentation and writing is strong. Quality of data are strong. The data support the conclusions for the most part. The results are significant: Generalized seizures and SWDs were reduced when both Torc1 and 2 were inactivated but not when one was inactivated.

Weaknesses: One of the limitations is a somewhat small sample size. Another is that there was hippocampal expression. A third is that recordings of seizures were not continuous and different for each mouse. Another concern is they only measured layer II/III neurons.

---

## [Author Response]

We thank all three Reviewers and the editors for the time and effort they put in reading and critiquing the manuscript. Our revised manuscript includes new data and analyses that address the original concerns. These include, 1) a new Supplemental Figure characterizing Cre expression and cellular phenotypes in the hippocampus, 2) new tables that give a more comprehensive picture of the EEG recordings and statistical analyses, 3) addition of whole cell electrophysiology data, and 4) rewriting to ensure that we do not state that either mTORC1 or mTORC2 hyperactivation is sufficient to cause epilepsy. We discuss the issue of statistical power to detect reduction in generalized seizure rate in the responses below. These suggestions and additions have improved the paper and we hope they will raise both significance and strength of support for the conclusions.

**Reviewer #1 (Public Review):**
Hyperactivation of mTOR signaling causes epilepsy. It has long been assumed that this occurs through overactivation of mTORC1, since treatment with the mTORC1 inhibitor rapamycin suppresses seizures in multiple animal models. However, the recent finding that genetic inhibition of mTORC1 via Raptor deletion did not stop seizures while inhibition of mTORC2 did, challenged this view (Chen et al, Nat Med, 2019). In the present study, the authors tested whether mTORC1 or mTORC2 inhibition alone was sufficient to block the disease phenotypes in a model of somatic Pten loss-of-function (a negative regulator of mTOR). They found that inactivation of either mTORC1 or mTORC2 alone normalized brain pathology but did not prevent seizures, whereas dual inactivation of mTORC1 and mTORC2 prevented seizures. As the functions of mTORC1 versus mTORC2 in epilepsy remain unclear, this study provides important insight into the roles of mTORC1 and mTORC2 in epilepsy caused by Pten loss and adds to the emerging body of evidence supporting a role for both complexes in the disease development.Strengths:The animal models and the experimental design employed in this study allow for a direct comparison between the effects of mTORC1, mTORC2, and mTORC1/mTORC2 inactivation (i.e., same animal background, same strategy and timing of gene inactivation, same brain region, etc.). Additionally, the conclusions on brain epileptic activity are supported by analysis of multiple EEG parameters, including seizure frequencies, sharp wave discharges, interictal spiking, and total power analyses.Weaknesses:(1) The sample size of the study is small and does not allow for the assessment of whether mTORC1 or mTORC2 inactivation reduces seizure frequency or incidence. This is a limitation of the study.

We agree that this is a minor limitation of the present study, however, for several reasons we decided not to pursue this question by increasing the number of animals. First, we performed a power analysis of the existing data. This analysis showed that we would need to use 89 animals per group to detect a significant difference (0.8 Power, p= 0.05, Mann-Whitney test) in the frequency of generalized seizures in the Pten-Raptor group and 31 animals per group in the Pten-Rictor group versus Pten alone. It is simply not feasible to perform video-EEG monitoring on this many animals for a single study. Second, even if we did do enough experiments to detect a reduction in seizure frequency, it is clear that neither Rptor nor Rictor deletion provides the kind normalization in brain activity that we seek in a targeted treatment. Both Pten-Rptor and Pten-Rictor animals still have very frequent spike-wave events (Fig. 3D) and highly abnormal interictal EEGs (Fig. 4), suggesting that even if generalized seizures were reduced, epileptic brain activity persists. This is in contrast to the triple KO animals, which have no increase in SWD above control level and very normal interictal EEG.

(2) The authors describe that they inactivated mTORC1 and mTORC2 in a new model of somatic Pten loss-of-function in the cortex. This is slightly misleading since Cre expression was found both in the cortex and the underlying hippocampus, as shown in Figure 1. Throughout the manuscript, they provide supporting histological data from the cortex. However, since Pten loss-of-function in the hippocampus can lead to hippocampal overgrowth and seizures, data showing the impact of the genetic rescue in the hippocampus would further strengthen the claim that neither mTORC1 nor mTORC2 inactivation prevents seizures.

Thank you for pointing out this issue. Cre expression was observed in both the cortex and the dorsal hippocampus in most animals, and we agree that differences in cortical versus hippocampal mTOR signaling could have differential contributions to epilepsy. We initially focused our studies on the cortex because spike-and-wave discharge, the most frequent and fully penetrant EEG phenotype in our model, is associated with cortical dysfunction. In our revised submission we have included a new Figure that quantifies Cre expression in the hippocampal subfields, as well as pS6, pAkt and soma size. These new data show that the amount of Cre expression in the hippocampus is not related to the occurrence of generalized seizures. The pattern of cell size changes in hippocampal neurons is the same as observed in cortical neurons. The levels of pS6 and pAkt are not much changed in the hippocampus, likely due to the sparse Cre expression there. We interpret these findings as supporting the conclusion that the reason we do not see seizure prevention by mTORC1 or mTORC2 inactivation is not due to hippocampal-specific dysfunction.

(3)Some of the methods for the EEG seizure analysis are unclear. The authors describe that for control and Pten-Raptor-Rictor LOF animals, all 10-second epochs in which signal amplitude exceeded 400 μV at two time-points at least 1 second apart were manually reviewed, whereas, for the Pten LOF, Pten-Raptor LOF, and Pten-Rictor LOF animals, at least 100 of the highest- amplitude traces were manually reviewed. Does this mean that not all flagged epochs were reviewed? This could potentially lead to missed seizures.

We reviewed at least 48 hours of data from each animal manually. All seizures that were identified during manual review were also identified by the automated detection program. It is possible but unlikely that there are missed seizures in the remaining data. We have added these details to the Methods of the revised submission.

(4) Additionally, the inclusion of how many consecutive hours were recorded among the ~150 hours of recording per animal would help readers with the interpretation of the data.

Thank you for this recommendation. Our revised submission includes a table with more information about the EEG recordings in the revised version of the manuscript. The number of consecutive hours recorded varied because the wireless system depends on battery life, which was inconsistent, but each animal was recorded for at least 48 consecutive hours on at least two occasions.

(5) Finally, it is surprising that mTORC2 inactivation completely rescued cortical thickness since such pathological phenotypes are thought to be conserved down the mTORC1 pathway. Additional comments on these findings in the Discussion would be interesting and useful to the readers.

We agree that the relationship between mTORC2, cortical thickness, and growth in general is an interesting topic with conflicting results in the literature. We didn’t add anything to the Discussion along these lines because we are up against word limits, but comment here that soma size was increased 120% by Pten inactivation and partially normalized to a 60% increase from Controls by mTORC2 inactivation (Fig. 2C). We and others have previously shown that mTORC2 inactivation (Rictor deletion) in neurons reduces brain size, neuron soma size, and dendritic outgrowth (PMIDs: 36526374, 32125271, 23569215). In our revised submission we also include new data showing that the membrane capacitance of Pten-Ric LOF neurons is normal. Thus, we do not find it completely surprising that mTORC2 inactivation reduces the cortical thickness increase caused by Pten loss. There may still be a slight increase in cortical thickness in Pten-Rictor animals, but it is statistically indistinguishable from Controls.

**Reviewer #2 (Public Review):**
Summary:The study by Cullen et al presents intriguing data regarding the contribution of mTOR complex 1 (mTORC1) versus mTORC2 or both in Pten-null-induced macrocephaly and epileptiform activity. The role of mTORC2 in mTORopathies, and in particular Pten loss-off-function (LOF)-induced pathology and seizures, is understudied and controversial. In addition, recent data provided evidence against the role of mTORC1 in PtenLOF-induced seizures. To address these controversies and the contribution of these mTOR complexes in PtenLOF-induced pathology and seizures, the authors injected a AAV9-Cre into the cortex of conditional single, double, and triple transgenic mice at postnatal day 0 to remove Pten, Pten+Raptor or Rictor, and Pten+raptor+rictor. Raptor and Rictor are essentially binding partners of mTORC1 and mTORC2, respectively. One major finding is that despite preventing mild macrocephaly and increased cell size, Raptor knockout (KO, decreased mTORC1 activity) did not prevent the occurrence of seizures and the rate of SWD event, and aggravated seizure duration. Similarly, Rictor KO (decreased mTORC2 activity) partially prevented mild macrocephaly and increased cell size but did not prevent the occurrence of seizures and did not affect seizure duration. However, Rictor KO reduced the rate of SWD events. Finally, the pathology and seizure/SWD activity were fully prevented in the double KO. These data suggest the contribution of both increased mTORC1 and mTORC2 in the pathology and epileptic activity of Pten LOF mice, emphasizing the importance of blocking both complexes for seizure treatment. Whether these data apply to other mTORopathies due to Tsc1, Tsc2, mTOR, AKT or other gene variants remains to be examined.Strengths:The strengths are as follows: 1) they address an important and controversial question that has clinical application, 2) the study uses a reliable and relatively easy method to KO specific genes in cortical neurons, based on AAV9 injections in pups. 2) they perform careful video-EEG analyses correlated with some aspects of cellular pathology.Weaknesses:The study has nevertheless a few weaknesses: 1) the conclusions are perhaps a bit overstated. The data do not show that increased mTORC1 or mTORC2 are sufficient to cause epilepsy.However the data clearly show that both increased mTORC1 and mTORC2 activity contribute to the pathology and seizure activity and as such are necessary for seizures to occur.

We agree that our findings do not directly show that either mTORC1 or mTORC2 hyperactivity are sufficient to cause seizures, as we do not individually hyperactivate each complex in the absence of any other manipulation. We interpreted our findings in this model as suggesting that either is sufficient based on the result that there is no epileptic activity when both are inactivated, and thus assume that there is not a third, mTOR-independent, mechanism that is contributing to epilepsy in Pten, Pten-Raptor, and Pten-Rictor animals. In addition, the histological data show that Raptor and Rictor loss each normalize activity through mTORC1 and mTORC2 respectively, suggesting that one in the absence of the other is sufficient.However, we agree that there could be other potential mTOR-independent pathways downstream of Pten loss that contribute to epilepsy. We have revised the manuscript to reflect this.

(2) The data related to the EEG would benefit from having more mice. Adding more mice would have helped determine whether there was a decrease in seizure activity with the Rictor or Raptor KO.

Please see response to Reviewer 1’s first Weakness.

(3) It would have been interesting to examine the impact of mTORC2 and mTORC1 overexpression related to point #1 above.

We are not sure that overexpression of individual components of mTORC1 or mTORC2 would result in their hyperactivation or lead to increases in downstream signaling. We believe that cleanly and directly hyperactivating mTORC1 or especially mTORC2 in vivo without affecting the other complex or other potential interacting pathways is a difficult task. Previous studies have used mTOR gain-of-function mutations as a means to selectively activate mTORC1 or pharmacological agents to selectively activate mTORC2, but it not clear to us that the former does not affect mTORC2 activity as well, or that the latter achieves activation of mTORC2 targets other than p-Akt 473, or that it is truly selective. We agree that these would be key experiments to further test the sufficiency hypothesis, but that the amount of work that would be required to perform them is more that what we can do in this Short Report.

**Reviewer #3 (Public Review):**
Summary: This study investigated the role of mTORC1 and 2 in a mouse model of developmental epilepsy which simulates epilepsy in cortical malformations. Given activation of genes such as PTEN activates TORC1, and this is considered to be excessive in cortical malformations, the authors asked whether inactivating mTORC1 and 2 would ameliorate the seizures and malformation in the mouse model. The work is highly significant because a new mouse model is used where Raptor and Rictor, which regulate mTORC1 and 2 respectively, were inactivated in one hemisphere of the cortex. The work is also significant because the deletion of both Raptor and Rictor improved the epilepsy and malformation. In the mouse model, the seizures were generalized or there were spike-wave discharges (SWD). They also examined the interictal EEG. The malformation was manifested by increased cortical thickness and soma size.Strengths: The presentation and writing are strong. The quality of data is strong. The data support the conclusions for the most part. The results are significant: Generalized seizures and SWDs were reduced when both Torc1 and 2 were inactivated but not when one was inactivated.Weaknesses: One of the limitations is that it is not clear whether the area of cortex where Raptor or Rictor were affected was the same in each animal.

Our revised submission includes new data showing that the area of affected cortex and hippocampus are similar across groups. (Figure 1A and Supplementary Figure 1)

Also, it is not clear which cortical cells were measured for soma size.

Soma size was measured by dividing Nissl stain images into a 10 mm2 grid. The somas of all GFP-expressing cells fully within three randomly selected grid squares in Layer II/III were manually traced. Three sections per animal at approximately Bregma -1.6, -2,1, and -2.6 were used. As Cre expression was driven by the hSyn promoter these cells include both excitatory and inhibitory cortical neurons.

Another limitation is that the hippocampus was affected as well as the cortex. One does not know the role of cortex vs. hippocampus. Any discussion about that would be good to add.

See response to Reviewer 1’s second Weakness.

It would also be useful to know if Raptor and Rictor are in glia, blood vessels, etc.

Raptor and Rictor are thought to be ubiquitously active in mammalian cells including glia and endothelial cells. Previous studies have shown that mTOR manipulation can aﬀect astrocyte function and blood vessel organization, however, our study induced gene knockout using an AAV that expressed Cre under control of the hSyn promoter, which has previously been shown to be selective for neurons. Manual assessment of Cre expression compared with DAPI, NeuN, and GFAP stains suggested that only neurons were aﬀected.

Recommendations for the authors: please note that you control which revisions to undertake from the public reviews and recommendations for the authors

**Reviewer #1 (Recommendations For The Authors):**
In addition to the comments in the public review, it is recommended that the authors provide a more representative ﬁgure for p-Akt staining in the Pten LOF condition in Figure 1 D2. The current ﬁgure is not convincing.

Thanks for the suggestion. We have replaced the images with zoomed in panels that beter demonstrate the diﬀerence.

Additionally, in the last paragraph of the discussion, there is a reference error to an incorrectpaper (reference 18) that should be corrected.

Thanks, corrected.

**Reviewer #2 (Recommendations For The Authors):**
Major comments:Comment 1: Some statements need clariﬁcations or changes.(1) Abstract: "spontaneous seizures and epileptiform activity persisted despite mTORC1 or mTORC2 inactivation alone but inactivating both mTORC1 and mTORC2 normalized pathology." Did inactivation of one only also normalized the pathology? Did inactivating both normalized the seizures? Pathology is not equal to seizures.

We have altered this statement to avoid ambiguity.

(2) Abstract: "These results suggest that hyperactivity of both mTORC1 and mTORC2 are suﬃcient to cause epilepsy,". Based on the abstract, it is not clear that it is suﬃcient. It is necessary.

We have altered this statement by removing the term “suﬃcient.”

(3) "Thus, there is strong evidence that hyperactivation of mTORC1 downstream of PTEN disruption causes the macrocephaly, epilepsy, early mortality, and synaptic dysregulation observed in humans and model organisms [17]" I would suggest adding that the strongest evidence is that mTOR GOF mutations lead to the same pathology and epilepsy, suggesting mTORC1 is suﬃcient. The other ﬁndings suggest that it is necessary.

Unless we misunderstand the Reviewer’s point, we believe this viewpoint is already encompassed by the proceeding text that “These phenotypes resemble those observed in models of mTORC1- speciﬁc hyperactivation.”

(4) Introduction (end): "suggesting that hyperactivity of either complex can lead to neuronal hyperexcitability and epilepsy".Comment 2: I do not agree with the title based on comment 1 above. You did not provide evidence that the mTORCs cause seizures. Your data suggest that they are necessary for seizures or contribute to seizures, but there is no evidence that mTORC2 can induce seizure.

We softened the title by replacing “cause” with “mediate.”

Comment 3: Fig. 1B. Could you beter describe the aﬀected regions. I can see other regions than just the cortex and hippocampus.

Almost all aﬀected cell bodies were in the cortex and hippocampus. The virus in the image is cell-ﬁlling and as such projections from aﬀected neurons throughout the brain can also be seen. We have clariﬁed this in the ﬁgure legend.

Comment 4: I feel unease about the number of animals recorded for EEG to assess seizure frequency. There is not enough power to draw clear conclusions. So, please make sure to not oversell your ﬁndings since it is all-or-nothing data (seizure or no seizure) in this case and the seizure frequency could very well be decreased with single mTOR LOF, but it is impossible to conclude. Maybe discuss this limitation of your study.

We have addressed this in the public comments response.

Minor:(1) Pten LOF: deﬁne the abbreviation.

Done

(2) Make sure that gene name in mice are not capitalized and italicized.

OK

(3) Fig 1C: could you specify in the results where the analysis was done.

Detail added to Methods (to keep Results concise for word limit)

(4) In the subtitle: "Concurrent mTORC1/2 inactivation, but neither alone, rescues epilepsy and interictal EEG abnormalities in focal Pten LOF". Replace "rescues" but prevents. This is not a rescue experiment since the LOF is done at the same time.

OK

(5) "GS did not appear to be correlated with mTOR pathway activity (Supplementary Figure 2)." Please can you do proper correlation analysis, by plotting all the values as a function of seizure frequency independent of the condition? There is also no correlation between pAKt and seizures.

Here are those data in Author response image 1. They are now part of Supplementary Figure 2.

**Author response image 1. sa4fig1:** 

**Reviewer #3 (Recommendations For The Authors):**
Figures 1 D, and E show images that are too small to judge. Where are the layers? Please add marks.

We replaced these images with larger zoomed in images to show group differences more clearly. The images no longer show multiple differentiable cortical layers.

If Fig 1 characterizes the model, where is the seizure data? When did they start? Where did they start? Was the focus of the cortical area aﬀected by PTEN loss of function?

Updated figure name to reflect content. Information about the seizure phenotypes is included in Figure 3.

Figure 2 The font size for the calibration is too small. The correlations are hard to see. Colors arenot easy to discriminate.

We edited the figure to correct these problems.

Figure 3 shows a clear eﬀect on generalized seizures but the text of the Results does not reﬂect that.

We wanted to be cautious about interpreting these data based on the issue raised by other reviewers that they are underpowered to detect seizure reduction in the Pten-Raptor and Pten-Rictor groups. We have updated the language to atempt to strike a beter balance between over- and under-interpretation. We also performed an additional analysis of the occurrence of generalized seizures to emphasize that only Control and PtRapRic animals have signiﬁcantly lower seizure occurrence that Pten LOF mice (Fig 3C).

For interictal power, was the same behavioral state chosen? Was a particular band aﬀected?

Epochs to be analyzed were selected automatically and were agnostic to behavioral state. Band-speciﬁc eﬀects are outlined in Figure 4B and Table [2].

There is no information about whether the model exhibits altered sleep, food intake, weight, etc.

We didn’t collect information on food intake. It would be possible to look at sleep from the EEG, but that is not something that we are prepared to do at this point. Weight at endpoint was not diﬀerent between genotypes but we did not collect longitudinal data on weight.

Were the sexes diﬀerent?

Included in new Table [1]

Where were EEG electrodes and were they subdural or not?

Additional detail on this has been added to Methods. The screws are placed in the skull but above the dura.

How long were continuous EEG records- the method just says 150 hr. per mouse in total.

Included in new Table [1]

The statistics don't discuss power, normality, whether variance was checked to ensure it did not diﬀer signiﬁcantly between groups, or whether data are mean +- sem or sd. For ANOVAs, were there multifactorial comparisons and what were F, df, and p values? Exact p for post hoc tests?

We have added a new table (Table [3]) that gives information on the exact test used, F, p values, and exact p for post hoc tests. Information regarding power, normality, variance, post- tests and multiple comparisons corrections have been added to Methods section “Statistical Analysis.”